# Targeting the cochlin/SFRP1/CaMKII axis in the ocular posterior pole prevents the progression of nonpathologic myopia

Chao Geng[1,4], Siyi Liu [1,4], Jindan Wang[1,4], Sennan Wang[1,4], Weiran Zhang[1,4], Hua Rong[1], Yunshan Cao[2], Shuqing Wang[3], Zhiqing Li[1] & Yan Zhang [1✉]

Myopia is a major public health issue. However, interventional modalities for nonpathologic myopia are limited due to its complicated pathogenesis and the lack of precise targets. Here, we show that in guinea pig form-deprived myopia (FDM) and lens-induced myopia (LIM) models, the early initiation, phenotypic correlation, and stable maintenance of cochlin protein upregulation at the interface between retinal photoreceptors and retinal pigment epithelium (RPE) is identified by a proteomic analysis of ocular posterior pole tissues. Then, a microarray analysis reveals that cochlin upregulates the expression of the secreted frizzled-related protein 1 (SFRP1) gene in human RPE cells. Moreover, SFRP-1 elevates the intracellular $Ca^{2+}$ concentration and activates $Ca^{2+}$/calmodulin-dependent protein kinase II (CaMKII) signaling in a simian choroidal vascular endothelial cell line, and elicits vascular endothelial cell dysfunction. Furthermore, genetic knockdown of the cochlin gene and pharmacological blockade of SFRP1 abrogates the reduced choroidal blood perfusion and prevents myopia progression in the FDM model. Collectively, this study identifies a novel signaling axis that may involve cochlin in the retina, SFRP1 in the RPE, and CaMKII in choroidal vascular endothelial cells and contribute to the pathogenesis of nonpathologic myopia, implicating the potential of cochlin and SFRP1 as myopia interventional targets.

[1] Tianjin Key Laboratory of Retinal Functions and Diseases, Tianjin Branch of National Clinical Research Center for Ocular Disease, Eye Institute and School of Optometry, Tianjin Medical University Eye Hospital, 300384 Tianjin, China. [2] Department of Cardiology, Gansu Provincial Hospital, Lanzhou University, 730000 Lanzhou, Gansu Province, China. [3] School of Pharmacy, Tianjin Medical University, 300070 Tianjin, China. [4] These authors contributed equally: Chao Geng, Siyi Liu, Jindan Wang, Sennan Wang, Weiran Zhang. ✉email: yanzhang04@tmu.edu.cn

Myopia has become a major public health issue, particularly for children and adolescents. In Asia, the prevalence of juvenile myopia is 80–90%, among which high myopia (HM, refractive errors >−6.00 diopters) accounts for 10–20%[1]. Moreover, nearly 50% of the world's population is predicted to suffer from myopia by 2050, with 10% of them being high myopia[2]. Without effective control, myopia will progress rapidly to HM, which may cause irreversible pathological lesions in the fundus[3]; if this occurs, the myopia is termed pathologic myopia (PM). Although medicinal agents and surgical procedures are available to combat PM, their therapeutic effects remain unsatisfactory partly due to the irreversible nature of PM-associated pathologies, hence patients with PM suffer from severe vision impairments or even legal blindness[4]. Since PM progresses from nonpathologic myopia, and most juvenile myopia belongs to nonpathologic myopia[5], it would be important to find interventional modalities to slow down or halt the progression of non-pathologic myopia, particularly in children and adolescents whose vision may be recovered by refractive surgeries after growth if no irreversible pathology is formed in the fundus. However, the current interventional modalities for nonpathologic myopia, such as wearing spectacles[6] or orthokeratology lenses[7], administering low-dose atropine[8], and laser-based refractive surgeries[9], are essentially symptomatic treatments, with the possibility of incurring side effects. This scenario indicates the complexity of myopia pathogenesis and the current paucity of precise molecular targets for myopic intervention.

Based on several theories, such as retinal local control[10], defocus[11], and scleral hypoxia theories[12], that have been proposed for myopia pathogenesis as well as the clinical observations of young subjects with nonpathologic myopia[13], researchers in the field began to realize that blurred images in the retina and waned blood perfusion in choroidal vessels are crucial to the incidence of nonpathological myopia[14,15]. However, the signaling axis that responds to blurred images in the retina and then leads to reduced choroidal blood perfusion remains unknown. To ascertain this signaling axis, we analyzed the ocular posterior pole tissues of guinea pig lens-induced myopia (LIM) and form-deprived myopia (FDM) models using high-throughput proteomics, and identified cochlin, the protein molecule with the most dramatic upregulation and the greatest statistical significance. Cochlin is a secretory extracellular matrix (ECM) protein encoded by the COCH gene and contains 550 amino acids[16]. Bhattacharya and colleagues reported that this ECM protein exhibited a time-dependent upregulation in the trabecular meshwork (TM) of both a mouse model of chronic glaucoma[17] and patients with primary open-angle glaucoma[18], resulting in age-related cochlin deposits and collagen diminution in the TM[18]. These reports indicate the possible pathogenicity of cochlin in eye disease. Furthermore, our immunohistochemical results revealed its location in retinal photoreceptors and its proximity to the retinal pigment epithelium (RPE); therefore, we speculate that cochlin might serve as a retinal molecular cue in response to the blurred image sensed by the photoreceptors and relays myopigenic information to the RPE.

Then, how is myopigenic information relayed from the RPE to the choroid vessels? A microarray analysis of cochlin-stimulated human RPE cells provided a clue and revealed the most upregulated expression of the secreted Frizzled-related protein 1 (SFRP-1) gene, which encodes a secretory glycoprotein with a molecular weight of 32.5 kD[19]. SFRP-1 belongs to the SFRP family and has a cysteine-rich domain homologous to the ligand binding site on Wnt receptors[20]. Moreover, the SFRP-1 gene was identified to be related to

myopia in 2 large-scale genome-wide association studies[21,22], implicating its clinical relevance to myopia formation. Additionally, from an anatomical perspective, the increased amount of SFRP-1, acting as the RPE-derived signal, would be able to diffuse along the concentration gradient and through the porous fibers of Bruch's membrane and fenestrated choriocapillaris to access choroidal vascular endothelial cells[23]. Although SFRP-1 is generally considered an inhibitor of the canonical Wnt signaling pathway, it has been implicated in the activation of the noncanonical Wnt signaling pathway by mobilizing $Ca^{2+}$/calmodulin-dependent protein kinase II (CaMKII) in vascular endothelial cells of breast cancer[24]. To this end, we hypothesize that blurred images on the retina could induce increased production of a photoreceptor ECM protein, cochlin, which may interact with RPE to boost SFRP-1 secretion. SFRP-1 might then activate nonconical Wnt $Ca^{2+}$/CaMKII signaling and cause dysfunction in choroidal vascular endothelial cells, leading to reduced choroidal blood perfusion and myopia formation. We sought to prove this hypothesis by a series of in vitro and in vivo gene expression analyses and functional assays, in the hope of shedding light on the pathogenesis of nonpathologic myopia. Furthermore, genetic knockdown of Coch gene expression and pharmacological blockade of SFRP-1 impeded the progression of nonpathologic myopia by augmenting choroidal blood perfusion in the FDM model, implicating the potential of these 2 molecules as myopia intervention targets.

## Results

**Phenotypes of FDM and LIM models: refraction, axial length, and choroidal and scleral thicknesses.** Prior to myopic induction, no difference in refraction of the right eye was found between the normal control, LIM, and FDM groups (all $P > 0.05$, Fig. 1a). At 2 w, 4 w, and 6 w following FDM induction, the right eye refraction in the FDM group was decreased 110.17%, 170.62%, and 211.14%, respectively, compared to that before induction, showing a trend of time-dependent reduction (Fig. 1a). Furthermore, the right eye refraction in the FDM group was significantly decreased in comparison to that of normal controls at each time point (all $P < 0.01$, NOR vs FDM, at 2, 4, 6 w, Fig. 1a). On the other hand, the changes in axial length (AL) paralleled those in refraction in this myopia model, demonstrating a time-dependent AL elongation (all $P < 0.001$, for FDM group, 0 w vs 2 w, 0 w vs 4 w, 0 w vs 6 w, Fig. 1c) and a greater AL in myopic animals than normal animals (all $P < 0.001$, NOR vs FDM, at 2, 4, 6 w, Fig. 1c). In contrast, the corneal curvature of the right eye did not change significantly at any time point following FDM induction (all $P > 0.05$, NOR vs FDM, at 2, 4, 6 w, Fig. 1e).

The LIM model recapitulated the biometric changes in the treated eye of the FDM model, although to a lesser extent at some time points in refraction and AL (for refraction $P < 0.05$; for AL, $P < 0.01$, FDM vs LIM at 4 w, Fig. 1a, c). These results demonstrated comparable phenotypes between the 2 myopia models and indicated that the induced myopia in this study was predominantly, if not completely, caused by AL elongation.

The differences between the treated (right) eye and the contralateral control (left) eye were also employed to quantify the refraction and AL. The trends of changes and statistical significance were similar to those using the data of the right eye only (Fig. 1b, d).

Moreover, hematoxylin and eosin (HE) staining revealed that the thicknesses of the sclera and the choroid were significantly reduced in the FDM model, being 66.47% and 57.55% of normal controls, at 6 weeks post myopic induction (both $P < 0.001$, NOR vs FDM, for both sclera and choroid, Fig. 1f, g). The scleral and

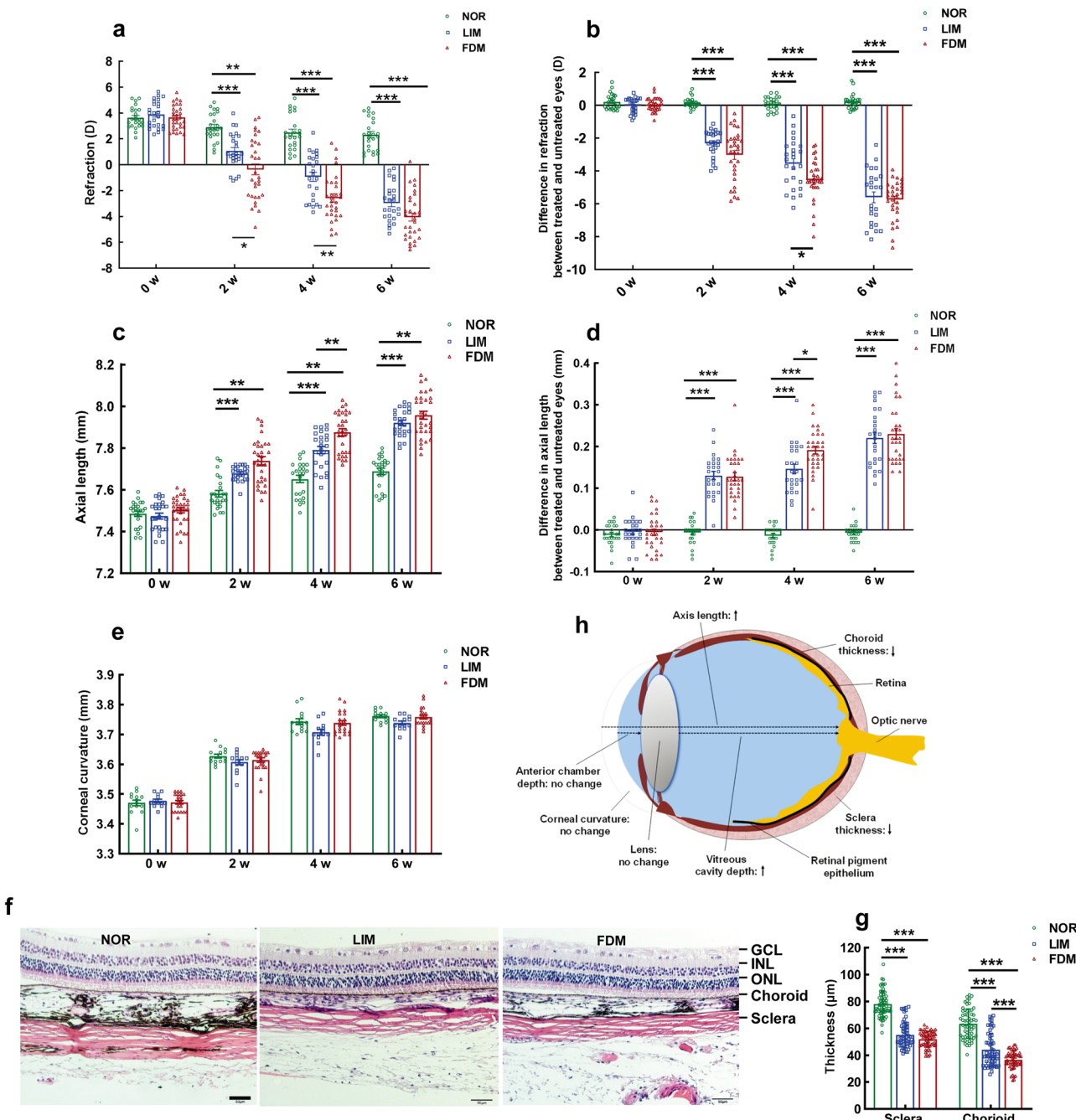

**Fig. 1 Characterization of guinea pig LIM and FDM models.** Using the data from the treated (right) eyes, the refraction (**a**), AL (**c**), and corneal curvature (**e**) of the guinea pig LIM and FDM models were measured and compared with two-way ANOVA (refraction: $n = 24$–$31$; AL: $n = 24$–$31$; corneal curvature: $n = 14$–$21$). Using the differences between the treated (right) eyes and untreated (left) eyes, the refraction (**b**) and AL (**d**) of the guinea pig LIM and FDM models were demonstrated and compared with two-way ANOVA (refraction: $n = 24$–$31$; AL: $n = 24$–$31$). Representative HE staining images of ocular posterior poles from the normal control, LIM, and FDM groups are shown in **f**. The thicknesses of the choroid and sclera in these images were quantified in **g** and compared with two-way ANOVA ($n = 54$–$60$). The changes in the biometric parameters in the 2 myopia models are illustrated in **h**. Scale bar = 50 μm. All data represent the mean ± SEM. *$P < 0.05$, **$P < 0.01$, ***$P < 0.001$. NOR normal control, GCL ganglion cell layer, INL inner nuclear layer, ONL outer nuclear layer, FDM form-deprived myopia, LIM lens-induced myopia, AL axial length.

choroidal thicknesses in LIM animals exhibited similar trends to FDM counterparts ($P < 0.001$, NOR vs LIM, for both sclera and choroid, Fig. 1f, g), except that choroidal attenuation in the LIM was less dramatic ($P < 0.001$, LIM vs FDM, for choroid, Fig. 1g). Therefore, the phenotypes of both LIM and FDM, including refraction exacerbation, AL elongation, and choroidal and scleral attenuation (Fig. 1h), resembled the clinical features of non-pathologic myopia.

**Proteomic analysis of ocular posterior pole tissues in FDM and LIM models.** To identify the molecular cues in the retina responsible for myopia pathogenesis, proteomic analysis of ocular posterior pole tissues was performed in both FDM and LIM models. Initially, fold change (log₂ |fold change|> 1) was set as the only criterion for differentially expressed genes (DEGs). Compared to the normal controls, 114 and 147 DEGs were identified in the FDM and LIM groups, respectively, whereas 112 DEGs

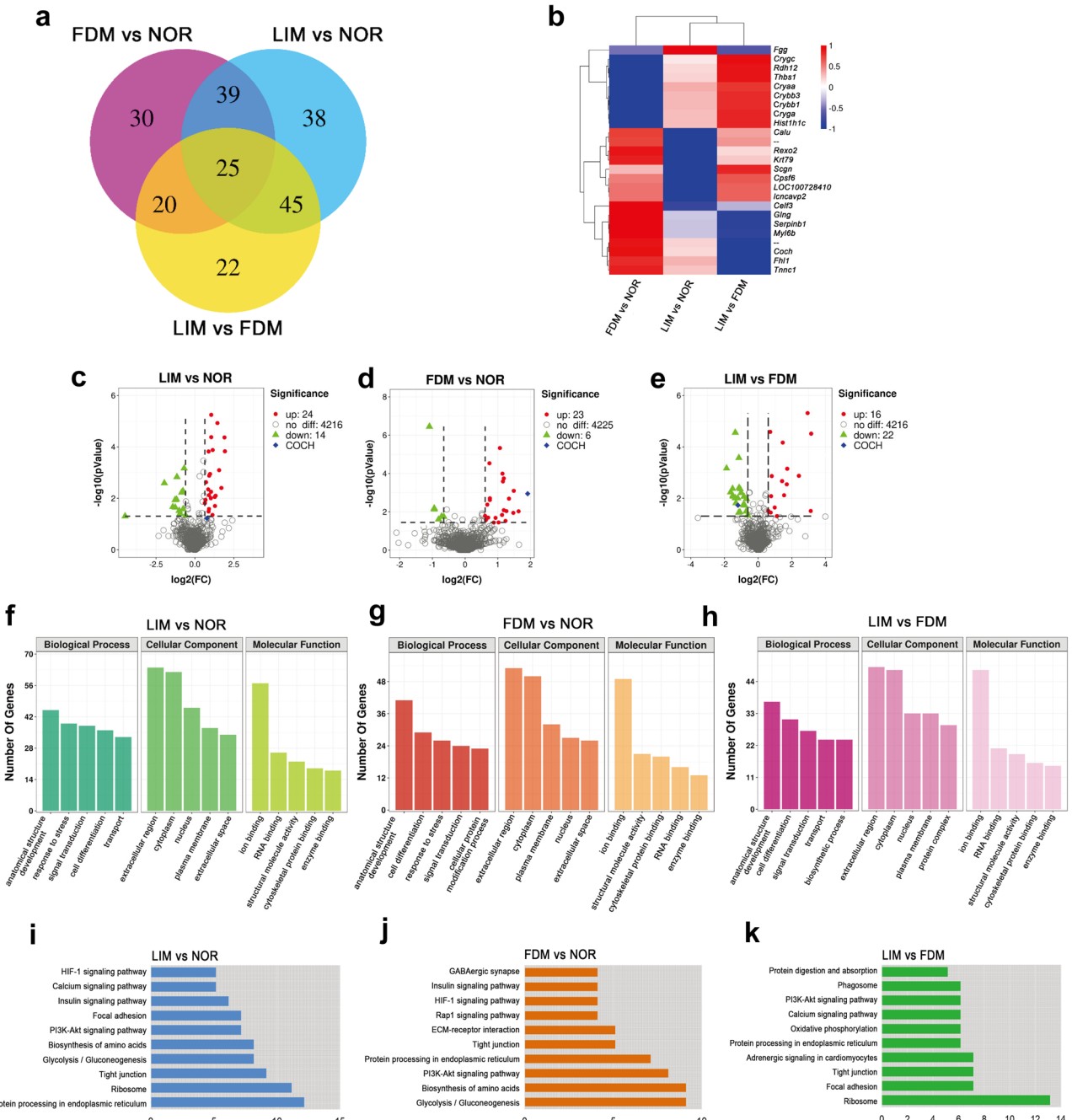

**Fig. 2 Proteomic analysis of ocular posterior pole tissues from LIM and FDM models.** Venn diagram of DEGs among experimental groups (**a**). Heatmap shows the differential protein expression profiling in FDM vs normal, LIM vs Normal, and LIM vs FDM groups (**b**). Under the premise of $P < 0.05$, the color scale is set according to the $log_2$ FC of the protein expression levels between the corresponding groups. Volcano maps of DEGs between LIM and normal control groups (**c**), FDM and normal control groups (**d**), and LIM and FDM groups (**e**), with red dots indicating upregulation, green triangle downregulation, gray circle no difference, and blue rhombus the position of cochlin. GO analyses of DEGs among experimental groups (**f–h**). KEGG analyses of DEGs among experimental groups (**i–k**). FC fold change of the protein expression levels, DEGs differentially expressed genes, GO Gene Ontology, KEGG Kyoto Encyclopedia of Genes and Genomes.

were found when the 2 myopic groups were compared with each other (Fig. 2a). Thereafter, both fold change and statistical significance ($log_2$ |fold change|> 1 and $P < 0.05$) were set as the criteria, and the expression patterns of the top 25 DEGs were displayed by a heatmap of hierarchical clustering analysis (Fig. 2b). Moreover, volcano plots showed 38 DEGs between the LIM and normal groups, among which 24 were upregulated and 14 were downregulated (Fig. 2c); 26 DEGs between the FDM and

normal groups, of which 23 were upregulated and 3 were downregulated (Fig. 2d); and 38 DEGs between the LIM and FDM groups, of which 16 were upregulated and 22 were downregulated (Fig. 2e). It is worth mentioning that the *Coch* gene, encoding the cochlin protein, ranked 1st among the top 10 DEGs between the FDM and normal groups, with its protein abundance in FDM being 3.78-fold greater than that in normal controls (Supplementary Table 1). In addition, cochlin protein expression

was also upregulated in the LIM model compared with normal controls (Fig. 2c), and its trend of expression paralleled the myopia severity among the experimental groups (Figs. 1a, b and 2c, d). Therefore, cochlin was selected as the potential molecular cue in the retina for expression validation.

The pairwise comparison results of GO analysis were highly consistent with each other. Anatomical structure development, extracellular region, and ion binding were the most enriched factors in all the comparisons under the categories of biological process, cellular component, and molecular function, respectively (Fig. 2f, g, h), highlighting the importance of these factors in the initiation and progression of nonpathologic myopia. On the other hand, KEGG analysis demonstrated that the HIF-1 signaling pathway, ECM-receptor pathway, focal adhesion pathway, protein processing in endoplasmic reticulum, $Ca^{2+}$ signaling pathway, and PI3K-Akt pathway were enriched with DEGs in the comparisons between the myopic models and the normal controls (Fig. 2i, j), implicating the pathogenic and regulatory roles of these pathways in nonpathologic myopia. In addition, the PI3K-Akt and $Ca^{2+}$ signaling pathways and protein processing in the endoplasmic reticulum were enriched when the 2 myopic groups were compared (Fig. 2k).

**Validation of cochlin expression in retinas of FDM and LIM models**. The myopia-induced upregulation of cochlin expression in ocular posterior poles was validated by conventional approaches, including quantitative real-time PCR (qPCR), western blots, and immunohistochemistry (IHC). The upregulation of cochlin gene expression was observed at both the transcript and protein levels as early as 1 w following FDM induction (both $P < 0.05$, FDM vs NOR at 1 w, Fig. 3a–c and Supplementary Fig. 1). Moreover, time-dependent increases in cochlin mRNA and protein levels were also detected during the first 3 w of FDM modeling ($P < 0.01$, FDM vs NOR at 2 w, $P < 0.001$, FDM vs NOR at 3 w, Fig. 3a; $P < 0.05$, FDM vs NOR at 2 w, $P < 0.01$, FDM vs NOR at 3 w, Fig. 3c). Furthermore, the trends of cochlin protein expression were significantly correlated with the changes in refraction and AL of the FDM model during the initial 3 w of induction (Fig. 3d–i, all $P < 0.05$).

At the end of the experiment, which is 6 w following myopia induction, qPCR showed that the mRNA levels of the cochlin gene in posterior pole tissues of LIM and FDM models were 2.32- and 4.55-fold higher than normal controls, respectively ($P < 0.05$, LIM vs NOR; $P < 0.001$, FDM vs NOR; $P < 0.001$ LIM vs FDM, Fig. 3j). The significantly upregulated gene expression was translated to the protein level, with the abundance of cochlin in the LIM and FDM groups being 1.88- and 2.98-fold greater than that in the normal controls, respectively ($P < 0.01$, LIM vs NOR; $P < 0.001$, FDM vs NOR; $P < 0.01$, LIM vs FDM, Fig. 3k, l and Supplementary Fig. 1). The results of western blots validated the proteomic findings of cochlin upregulation in the myopic models and implicated the correlation of its abundancy with myopia severity.

Interestingly, IHC revealed that the cochlin protein was primarily localized in the outer nuclear layer, the inner segments, and the initial portion of the outer segments of retinal photoreceptors, with a sparse distribution in the RPE (Fig. 3m). The localization of the cochlin protein was essentially consistent with its reported function as an ECM protein[19,21]. The semiquantification of the IHC demonstrated that the immunostaining of the cochlin protein was significantly augmented in LIM and FDM retinas at 6 w post myopic induction (both $P < 0.001$, for LIM vs NOR and FDM vs NOR; $P < 0.05$, LIM vs FDM; Fig. 3n), and the trend of the changes in cochlin expression was consistent with that shown in the western blot analysis (Fig. 3l).

The immunofluorescence confirmed the localization (Supplementary Fig. 2a) and trend of expression of cochlin (Supplementary Fig. 2b, c), the staining of photoreceptor inner segments was more prominent (Supplementary Fig. 2b).

Taken together, the early initiation, phenotypic correlation, and stable maintenance of cochlin upregulation in myopia models validated the association of cochlin overexpression with myopia pathogenesis, implicating the role of cochlin as a molecular cue in the retina during myopigenesis.

**Transcriptomic analysis of cochlin-stimulated RPE cells and validation of SFRP1 expression**. We stimulated ARPE-19, a human RPE cell line, with oligomerized recombinant human cochlin protein. The stimulated cells and normal controls were then subjected to a microarray analysis to examine the effects of this ECM protein on RPE cells capable of producing a repertoire of signaling molecules (Fig. 4a). The hierarchical clustering analysis of the microarray data showed differential gene expression patterns of the cochlin-treated cells and their normal counterparts (Fig. 4b). The dot plot demonstrated that among the DEGs, 156 genes were upregulated and 321 were downregulated in the cochlin-stimulated ARPE-19 cells compared to the normal cells (Fig. 4c). Moreover, GO analysis suggested that cochlin stimulation may elicit biological processes such as apoptosis, lipoxygenase mobilization, and protein kinase C activation (Fig. 4d). The site of action may be the cell membrane, where the activities of transmembrane transporters and ion channels could be enhanced by stimulation (Fig. 4e, f). KEGG analysis revealed that the DEGs were enriched on the pathways of polyunsaturated fatty acid metabolism and ligand-receptor interaction (Fig. 4g). Notably, according to the microarray data, the *SFRP1* gene was the most upregulated among the DEGs, with its transcript abundance in cochlin-treated cells being 8.61-fold greater than that in normal controls (Supplementary Table 2). Given the most dramatic expression changes in response to cochlin stimulation and its reported role as a susceptible gene for myopia[21,22,25], *SFRP1* was selected as another potential target, and its expression in RPE cells was validated by conventional approaches. The qPCR results showed that *SFRP1* transcript levels were boosted 4.41- and 2.74-fold by cochlin stimulation in ARPE-19 cells and human primary RPE cells, respectively (both $P < 0.01$, Cochlin vs NOR, Fig. 4h, j). Additionally, enzyme-linked immunosorbent assay (ELISA) showed that the concentrations of SFRP1 protein in the conditioned media (CM) of cochlin-stimulated ARPE-19 cells and human primary RPE cells were 31.59 ng/ml and 33.33 ng/ml, respectively, which were 8.40- and 10.16-fold higher than the corresponding controls ($P < 0.05$ for ARPE-19, Cochlin vs NOR, Fig. 4i; $P < 0.01$ for human primary RPE, Cochlin vs NOR, Fig. 4k).

**SFRP-1 and conditioned media of cochlin-treated RPE cells caused dysfunction of choroidal vascular endothelial cells**. The effects of SFRP-1 (Fig. 5a) or the CM of cochlin-treated ARPE-19 cells (Supplementary Fig. 2a) on choroidal vascular endothelial cells were examined. SFRP-1 at 30 ng/ml, the concentration detected in the culture media of cochlin-stimulated RPE cells (Fig. 4i, k) or the CM were incubated with RF/6 A cells, a simian choroidal vascular endothelial cell line. A specific inhibitor of SFRP1, WAY 316606[26], and its vehicle control dimethyl sulfoxide (DMSO), as well as a specific activator of the canonical Wnt signaling pathway, Wnt3a[27], and its vehicle control phosphate-buffered saline (PBS), were also included in the culture media (Fig. 5a) or the CM (Supplementary Fig. 3a) of the corresponding experimental groups.

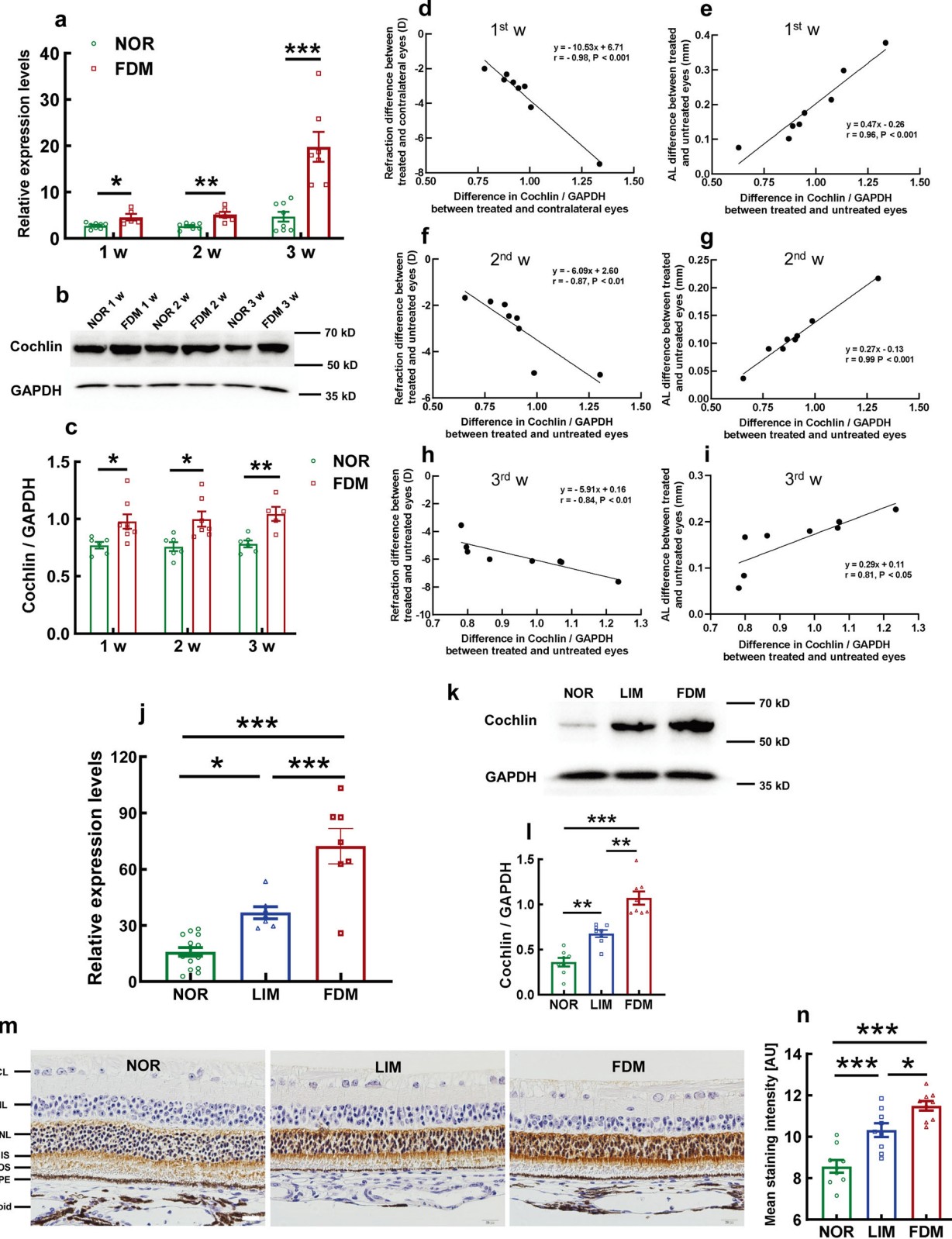

At 24 h post incubation, the Cell Counting Kit-8 (CCK-8) assay was performed to evaluate cell proliferation. The results showed no significant difference in absorbance at 450 nm (OD 450) among the experimental groups, indicating that SFRP1, at least at the concentration of 30 ng/ml, did not affect the proliferation of choroidal vascular endothelial cells (Fig. 5b and Supplementary Fig. 3b).

Caspase 3/7 activity was measured in RF/6A cells as one of the apoptotic assays. The normalized caspase 3/7 activity was boosted 39.75% by SFRP-1 compared to that of the normal controls ($P < 0.05$, SFRP-1 vs Normal, Fig. 5c). The increased caspase activity was restored to the normal level by WAY 316606 and Wnt3a but not by their vehicle controls ($P < 0.01$, for SFRP-1 vs SFRP-1 + WAY; both $P < 0.001$, for SFRP-1 vs SFRP-1 + Wnt3a,

**Fig. 3 Validation of cochlin gene and protein expression in FDM and LIM models.** The expression of the *Coch* gene in the FDM group during the first 3 w of myopia induction is shown in **a** and was compared with that in the normal control group using two-way ANOVA ($n = 5$–8). Representative blot of cochlin protein expression during the initial 3-w induction is shown in **b**. The blots were quantified in **c** and compared using two-way ANOVA ($n = 5$–9). The differences in cochlin protein levels between treated and untreated eyes were negatively correlated with the refraction differences between the 2 eyes during the 1st (**d**), 2nd (**f**), and 3rd (**h**) w of FDM modeling. The cochlin expression differences between treated and untreated eyes were positively correlated with the AL differences between the 2 eyes during the 1st (**e**), 2nd (**g**), and 3rd (**i**) w of FDM modeling. *Coch* gene expression levels in LIM and FDM models at 6 w post induction were shown in **j** and compared with those of the normal controls using one-way ANOVA ($n = 7$–14). Representative blots of the cochlin protein in LIM and FDM models at 6 w after myopic induction (**k**). The blots were quantified in **l** and compared using one-way ANOVA ($n = 8$). Representative pictures of cochlin immunohistochemistry staining in the normal control, LIM, and FDM groups at 6 w post myopia induction (**m**). Scale bar = 20 μm. The mean staining intensity was quantified in **n** and compared using one-way ANOVA ($n = 9$). All data represent the mean ± SEM. *$P < 0.05$, **$P < 0.01$, ***$P < 0.001$. NOR normal control, FDM form-deprived myopia, LIM lens-induced myopia, AL axial length, w- week, AU arbitrary units.

SFRP-1 + Wnt3a vs SFRP-1 + PBS; $P < 0.05$, SFRP-1 + WAY vs SFRP-1 + DMSO; Fig. 5c). Similar effects of the CM were observed in Supplementary Fig. 3c. The proapoptotic effect of the CM was corroborated by another assay, annexin V immunostaining followed by flow cytometry (all $P < 0.001$, CM vs Cochlin CM, Cochlin CM vs Cochlin CM + WAY, Cochlin CM vs Cochlin CM + Wnt3a, Cochlin CM + WAY vs Cochlin CM + DMSO, Cochlin CM + Wnt3a vs Cochlin CM + PBS, Supplementary Fig. 4).

In the Matrigel assay, the number of cell aggregates per image in the normal group decreased 27.25% after incubation with SFRP-1 ($P < 0.01$, normal vs SFRP-1, Fig. 5d, e). Such a reduction was restored by WAY 316606 and Wnt3a (both $P < 0.001$, SFRP-1 vs SFRP-1 + WAY, SFRP-1 vs SFRP-1 + Wnt3a; all $P > 0.05$, Normal vs SFRP-1 + WAY, Normal vs SFRP-1 + Wnt3a, Fig. 5d, e) but not by DMSO and PBS (both $P < 0.05$, normal vs SFRP-1 + DMSO, normal vs SFRP-1 + PBS; SFRP-1 + WAY vs SFRP-1 + DMSO; $P < 0.001$, SFRP-1 + Wnt3a vs SFRP-1 + PBS, Fig. 5d, e). The CM of cochlin-treated ARPE-19 cells had comparable effects on RF-6A cells (Supplementary Fig. 3d, e).

At 24 h following treatment with SFRP-1, the trend of cell migration was assessed by the transwell assay, demonstrating that the trends of migrated cells among the experimental groups paralleled those in the apoptotic assays. WAY 316606 and Wnt3a, but not DMSO and PBS, blocked the increase in the number of migrated RF-6 A cells induced by SFRP-1 or CM (all $P < 0.001$, normal vs SFRP-1, SFRP-1 vs SFRP-1 + WAY, SFRP-1 vs SFRP-1 + Wnt3a, SFRP-1 + WAY vs SFRP-1 + DMSO, SFRP-1 + Wnt3a vs SFRP-1 + PBS, Fig. 5f, g and Supplementary Fig. 3f, g).

Therefore, the results of these functional assays suggested that SFRP-1 per se or SFRP1-containing CM harvested from cochlin-stimulated ARPE-19 cells promoted apoptosis, inhibited cell aggregation, and enhanced the migration of RF/6 A cells, leading to dysfunction of choroidal endothelial cells, which could be abrogated by a specific antagonist of SFRP1 or an activator of the canonical Wnt signaling pathway.

SFRP-1 treatment of RF-6A cells induced an 18.08% increase in intracellular $Ca^{2+}$ concentration ($P < 0.01$, normal vs SFRP-1, Fig. 5h). The application of WAY 316606, but not its solvent DMSO, lowered the $Ca^{2+}$ concentration to the normal level ($P < 0.01$, SFRP-1 vs SFRP-1 + WAY; $P < 0.05$, SFRP-1 + WAY vs SFRP-1 + DMSO; Fig. 5h).

Based on the literature reports, when the canonical Wnt signaling is not activated, β-catenin is phosphorylated at serine 33, 37 (Ser 33, 37) and threonine 41 (Thr 41), promoting its degradation. By contrast, when the canonical Wnt signaling is activated by Wnt ligands, the phosphorylation at Ser 33, 37 and Thr 41 is inhibited, thus leading to stabilization and accumulation of β-catenin[28,29]. Therefore, the levels of phosphorylated β-catenin (p-β-catenin) at Ser 33, 37 and Thr 41 and the levels of β-catenin were analyzed by western blots in this study to evaluate the activity

of canonical Wnt signaling in choroidal vascular endothelial cells. The results showed that the levels of p-β-catenin and β-catenin did not vary significantly under the conditions including normal or conditioned culture media, SFRP-1 alone or with its antagonist or vehicle control (Fig. 5i–k and Supplementary Fig. 3h–j), suggesting that SFRP-1 at 30 ng/ml or the CM did not activate the canonical Wnt signaling. However, Wnt3a at 0.3 μg/ml, either added with SFRP-1 or to the CM, significantly reduced the levels of p-β-catenin at Ser 33,37 and Thr 41 and increased the abundance of β-catenin in RF/6 A cells (Fig. 5i–k, Supplementary Fig. 3h–j, and Supplementary Figs. 5 and 6; for p-β-catenin in Fig. 5j, all $P < 0.01$, SFRP-1 vs SFRP-1 + Wnt3a, SFRP-1 + WAY vs SFRP-1 + Wnt3a, SFRP-1 + DMSO vs SFRP-1 + Wnt3a; $P < 0.001$, SFRP-1 vs SFRP-1 + PBS; for β-catenin in Fig. 5k, all $P < 0.001$, Normal vs SFRP-1 + Wnt3a; SFRP-1 vs SFRP-1 + Wnt3a, SFRP-1 + WAY vs SFRP-1 + Wnt3a, SFRP-1 + DMSO vs SFRP-1 + Wnt3a, SFRP-1 + Wnt3a vs SFRP-1 + PBS), hence the ratio of p-β-catenin/β-catenin was dramatically reduced following the addition of Wnt3a (Fig. 5l and Supplementary Fig. 3k; all $P < 0.001$, Normal vs SFRP-1 + Wnt3a; SFRP-1 vs SFRP-1 + Wnt3a, SFRP-1 + WAY vs SFRP-1 + Wnt3a, SFRP-1 + DMSO vs SFRP-1 + Wnt3a, SFRP-1 + Wnt3a vs SFRP-1 + PBS). This experiment also served as a positive control and implicated that the canonical Wnt signaling was activatable in the RF/6 A cells but was not activated by SFRP-1 or the CM at least under our experimental paradigm.

Importantly, SFRP-1 significantly elevated the levels of phosphorylated CaMKII (p-CaMKII), including both α and β isoforms ($P < 0.001$, SFRP-1 vs normal, Fig. 5i, m and Supplementary Fig. 6). The aberrant elevation of p-CaMKII was abolished by both WAY 316606 and Wnt3a (both $P < 0.05$, SFRP-1 vs SFRP-1 + WAY, SFRP-1 vs SFRP-1 + Wnt3a) but not by their vehicle controls ($P < 0.01$, SFRP-1 + WAY vs SFRP-1 + DMSO; $P < 0.05$, SFRP-1 + Wnt3a vs SFRP-1 + PBS, Fig. 5i, m and Supplementary Fig. 6). The abundance of CaMKII exhibited small fluctuations, yet the difference among groups was far from reaching statistical significance (Fig. 5i, n and Supplementary Fig. 6). Therefore, the ratio of p-CaMKII/CaMKII also exhibited significant changes among the experimental groups (Fig. 5o). Comparable results were observed when the cells were treated with the CM from the cochlin-stimulated ARPE-19 cells (Supplementary Fig. 3h, l–n and Supplementary Fig. 5).

The results of the functional assays and cell signaling analyses imply that SFRP-1, either directly or in the CM might cause dysfunction of choroidal vascular endothelial cells through elevation of intracellular $Ca^{2+}$ concentration and activation of CaMKII, the 2 key steps in the noncanonical Wnt signaling pathway[30].

**Targeting the cochlin/SFRP1/CaMKII axis prevented myopia progression in the FDM model by increasing choroidal blood perfusion.** Cochlin and SFRP1 are the most important molecular

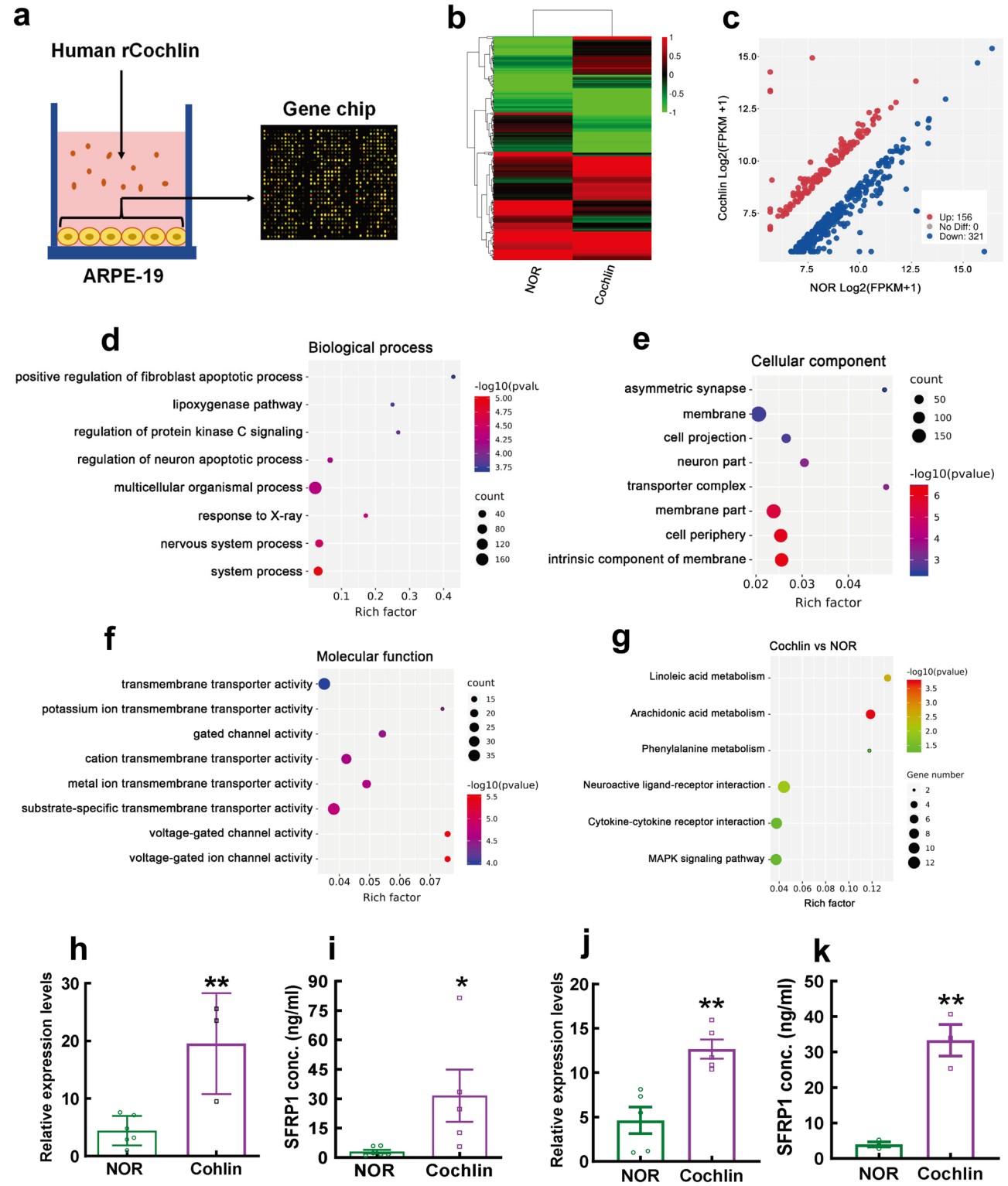

targets identified in this study. Genetic and pharmacological approaches were employed using a shRNA against the guinea pig *Coch* transcript and a specific inhibitor of SFRP1, respectively, to validate the effects of targeting these 2 molecules in FDM, the guinea pig model of nonpathologic myopia with more aggravated phenotypes in comparison to LIM.

The data from the right eyes of the experimental animals showed no significant difference in refraction prior to FDM induction (Fig. 6a). At 1 w post induction, the refraction in the

FDM group exhibited an 11.63-fold plummet from the norm ($P < 0.01$, NOR vs FDM, Fig. 6a). Such a dramatic reduction in refraction was prevented by an intravitreal injection of lentiviral particles carrying shRNA against the *Coch* gene or by 2 intravitreal injections of WAY 316606 ($P < 0.05$, shRNA vs FDM, WAY vs FDM, Fig. 6a). The preventive effects of the shRNA and the SFRP1 inhibitor were maintained until the end of the experiment at 6 w post induction ($P < 0.001$, shRNA vs FDM, $P < 0.01$, WAY vs FDM, Fig. 6a). The changes in refraction

**Fig. 4 Microarray analysis of RPE cells stimulated with recombinant human cochlin protein.** The rationale of this experiment is illustrated in **a**. Heatmap demonstrates the differential gene expression profiling of the ARPE-19 cells treated with or without cochlin (**b**). The color scale is set according to the $\log_2$ FC of the gene expression levels between Cochlin and NOR groups. Dot plot of the DEGs between the ARPE-19 cells treated with or without cochlin (**c**). GO analysis of the DEGs between the 2 groups of cells (**d**–**f**). KEGG analysis of the DEGs between the 2 groups of cells (**g**). Gene expression levels of *SFRP1* in ARPE-19 cells treated with or without cochlin were shown in **h** and compared with unpaired *t* test ($n = 3$–6). SFRP1 protein concentration (ng/ml) in the culture media of cochlin-treated ARPE-19 cells was shown in **i** and compared with that in the media of untreated ARPE-19 cells using unpaired *t* test ($n = 5$–6). *SFRP1* gene expression in human primary RPE cells treated with or without cochlin was demonstrated in **j** and compared using unpaired *t* test ($n = 5$). SFRP1 protein concentration (ng/ml) in the culture media of cochlin-treated human primary RPE cells was shown in **k** and compared with that of the normal control cells using unpaired *t* test ($n = 3$). All data represent the mean ± SEM. *$P < 0.05$, **$P < 0.01$. RPE retinal pigment epithelium, ARPE-19 a RPE cell line of human origin, DEGs differentially expressed genes, NOR normal control group, Cochlin cochlin-treated group, FC fold change of gene expression levels, GO Gene Ontology, KEGG Kyoto Encyclopedia of Genes and Genomes, SFRP1 secreted frizzled-related protein 1.

exhibited similar trends and statistical significance when the differences between the treated eye (right eye) and the contralateral eye (left eye) were used for analysis (Fig. 6c).

The AL of the right eyes did not differ among the groups before myopic induction (Fig. 6b). At 1 w post induction, the AL in the FDM group was significantly longer than that of normal controls ($P < 0.05$, NOR vs FDM, for 1 w, Fig. 6b). Intravitreal administration of *Coch* shRNA or WAY 316606 curtailed the length of the axis in FDM animals; however, the differences in AL did not reach statistical significance at this time point ($P > 0.05$, FDM vs shRNA, FDM vs WAY 316606 for 1 w, Fig. 6b). At 6 w following the induction, compared to the normal controls, the AL in the FDM group extended 102.71% ($P < 0.001$, FDM vs NOR for 6 w, Fig. 6b), which was precluded by intravitreal administration of either the shRNA or WAY 316606 ($P < 0.05$, FDM vs shRNA, FDM vs WAY 316606 for 6 w, Fig. 6b). When the bilateral AL differences were used for analysis, the trends and statistical significance among the experimental groups were comparable to those using the data of the right eyes (Fig. 6d).

OCTA was used to measure blood perfusion in choroidal vessels in live animals (Fig. 6e, f). A customized computer program was developed to analyze OCTA images and quantify blood flow signals in each image. The results demonstrated that the blood perfusion in choroidal vessels was reduced 27.30% and 16.65% at 1 w and 6 w following FDM induction, respectively, compared to the normal controls (both $P < 0.001$, NOR vs FDM at 1 w and 6 w, Fig. 6g, h). Such reductions were abrogated by intravitreal administration of the shRNA against the cochlin gene and a specific inhibitor of SFRP1 at both time points (all $P < 0.001$, FDM vs shRNA at 1 w and 6 w, FDM vs WAY at 1 w and 6 w, Fig. 6g, h). Nonetheless, the scramble (Scr) and DMSO controls did not exert such effects (all $P > 0.05$, FDM vs Scr at 1 w and 6 w, FDM vs DMSO at 1 w and 6 w; $P < 0.01$, shRNA vs Scr at 1 w; all $P < 0.001$, shRNA vs Scr at 6 w, WAY vs DMSO at 1 w and 6 w; Fig. 6g, h).

The hemodynamic conditions of choroidal vessels were examined by morphometric analysis of choroidal vessel lumen areas in guinea pig eyeball paraffin sections (Fig. 6i). The results showed similar intergroup trends of the lumen areas of choroidal vessels to those revealed by OCTA (both $P < 0.001$, NOR vs FDM at 1 w and 6 w; $P = 0.054$, FDM vs shRNA at 1 w; all $P < 0.001$, FDM vs shRNA at 6 w, FDM vs WAY at 1 w and 6 w; $P < 0.05$, shRNA vs Scr at 1 w; $P < 0.01$, WAY vs DMSO at 1 w; both $P < 0.001$, shRNA vs Scr at 6 w, WAY vs DMSO at 6 w; Fig. 6j, k), providing an anatomical basis for the changes in choroidal circulation when targeting cochlin and SFRP1 for early intervention of nonpathologic myopia.

**Molecular effects of intervening in the two actionable targets, cochlin, and SFRP1, in the FDM model.** Western blots showed that an intravitreal injection of shRNA, but not Scr, resulted in 39.75% and 63.72% reductions in cochlin protein abundance in

ocular posterior pole tissues at 1 w and 6 w following myopic induction, which subdued the aberrant upregulation of this protein induced by FDM modeling ($P < 0.001$, NOR vs FDM at 1 w; $P < 0.05$, NOR vs FDM at 6 w; $P < 0.01$, FDM vs shRNA at 1 w; $P < 0.05$, FDM vs shRNA at 6 w; both $P < 0.05$, shRNA vs Scr at 1 w and 6 w; Fig. 7a–d and Supplementary Fig. 8).

Furthermore, the 1-week and 6-week induction significantly elevated p-CaMKII levels in the ocular posterior poles of FDM animals, which was abolished by shRNA-mediated *Coch* gene knockdown and WAY 316606-mediated SFRP1 inhibition, thus normalizing the FDM-elicited overactivation of noncanonical Wnt signaling (both $P < 0.05$, NOR vs FDM at 1 w and 6 w; both $P < 0.01$, shRNA vs Scr at 1 w, WAY vs DMSO at 1 w; both $P < 0.001$, shRNA vs Scr at 6 w, WAY vs DMSO at 6 w; Fig. 7e, i, f, j and Supplementary Fig. 8). Unlike the choroidal vascular endothelial cells, CaMKII in the FDM animals exhibited similar significant changes to its phosphorylated counterpart (all $P < 0.05$, NOR vs FDM at 1 w, shRNA vs Scr at 1 w, WAY vs DMSO at 1 w and 6 w; $P < 0.01$, NOR vs FDM at 6 w; $P < 0.001$, shRNA vs Scr at 6 w; Fig. 7e, i, g, k and Supplementary Fig. 8), and the ratios of p-CaMKII/CaMKII did not reach statistical significance at either time point (all $P > 0.05$, Fig. 7h, l).

## Discussion

In reference to the previously proposed pathogenic theories of nonpathological myopia[10–12] and given that most signaling molecules, such as neurotransmitters and neuromodulators[31], are secretable, we speculated that the molecular cue initiating non-pathologic myopia could be generated, but not necessarily reside, in the retina. Furthermore, we hypothesized that the signals may be relayed sequentially in the ocular posterior pole between the major tissues: retina, RPE, and choriocapillaris and choroidal vessels, ultimately resulting in waned blood flow in the choroid. To prove this hypothesis, we established guinea pig LIM and FDM models, collected the ocular posterior pole tissues instead of the retina or the sclera samples used in other studies[32,33], and subjected them to a high-throughput proteomic analysis. Then, cochlin, an ECM protein, with the most prominent upregulation in the myopia models, stood out (Figs. 2 and 3 and Supplementary Table 1). The spatiotemporal characteristics and phenotypic correlation of cochlin expression supported its role as a retinal molecular cue relaying myopigenic information. Next, a microarray analysis revealed that the expression of another secretory protein, SFRP1, was the most dramatically upregulated in the cochlin-stimulated RPE cells (Fig. 4 and Supplementary Table 2). SFRP1, serving as a small-molecule signal derived from the RPE, elevated the levels of $Ca^{2+}$ and p-CaMKII, the 2 key molecules in the noncanonical Wnt signaling pathway, in the endothelial cells of the choriocapillaris and choroidal vessels (Fig. 5h, i, k) and caused dysfunction of the endothelial cells, such as promoted apoptosis, suppressed aggregation, and enhanced migration (Fig. 5c–g), leading to reduced choroidal blood

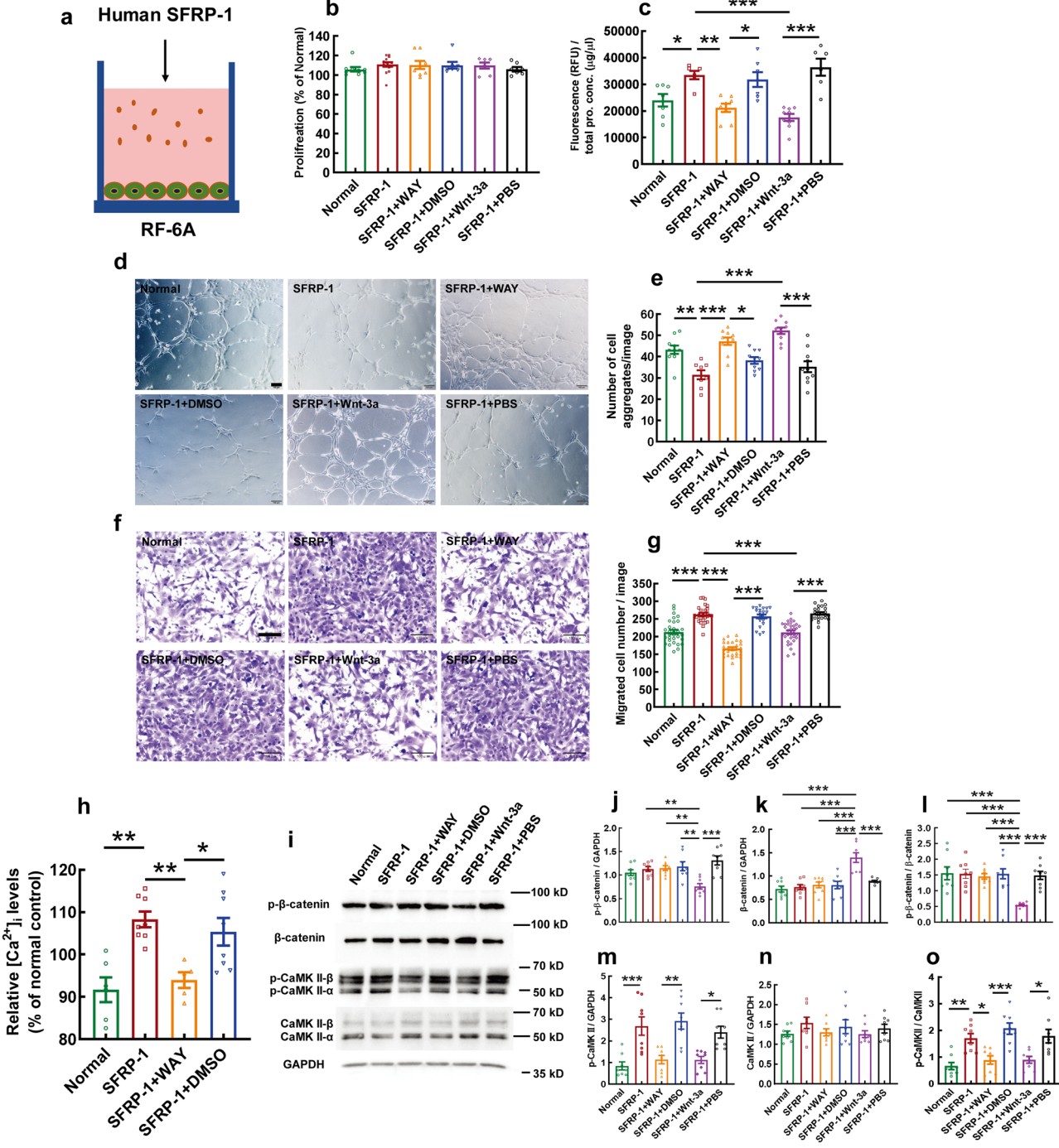

**Fig. 5 SFRP-1 caused dysfunction of choroidal vascular endothelial cells through activation of noncanonical Wnt/Ca$^{2+}$/CaMKII signaling.** The experimental rationale is illustrated in **a**. The effects of SFRP-1 on the proliferation of choroidal vascular endothelial cells were examined by CCK-8 assay (**b**) and compared using one-way ANOVA ($n = 7$–14). The effects of SFRP-1 on the apoptosis of choroidal vascular endothelial cells were examined by caspase 3/7 activity assay (**c**) and compared using one-way ANOVA ($n = 6$–9). Representative images of the Matrigel assay of SFRP-1-treated choroidal vascular endothelial cells (**d**). Scale bar = 100 μm. Quantification of the Matrigel assay (**e**) was compared with one-way ANOVA ($n = 10$). Representative images of the Transwell assay of SFRP-1-treated choroidal vascular endothelial cells (**f**). Scale bar = 100 μm. Quantification of the Transwell assay (**g**) was compared with one-way ANOVA ($n = 22$–33). Intracellular Ca$^{2+}$ concentration after SFRP-1 treatment (**h**). The relative intracellular Ca$^{2+}$ levels were compared using one-way ANOVA ($n = 5$–8). Representative western blots of p-β-catenin, β-catenin, p-CaMKII, and CaMKII in SFRP-1-treated choroidal vascular endothelial cells (**i**). Quantification of the western blots (**j**–**o**) was compared using one-way ANOVA ($n = 8$). All data represent the mean ± SEM. *$P < 0.05$, **$P < 0.01$, ***$P < 0.001$. SFRP-1 secreted frizzled-related protein 1, CCK Cell Counting Kit, p-CaMKII phosphorylated Ca$^{2+}$/calmodulin-dependent protein kinase II, CaMKII Ca$^{2+}$/calmodulin-dependent protein kinase II.

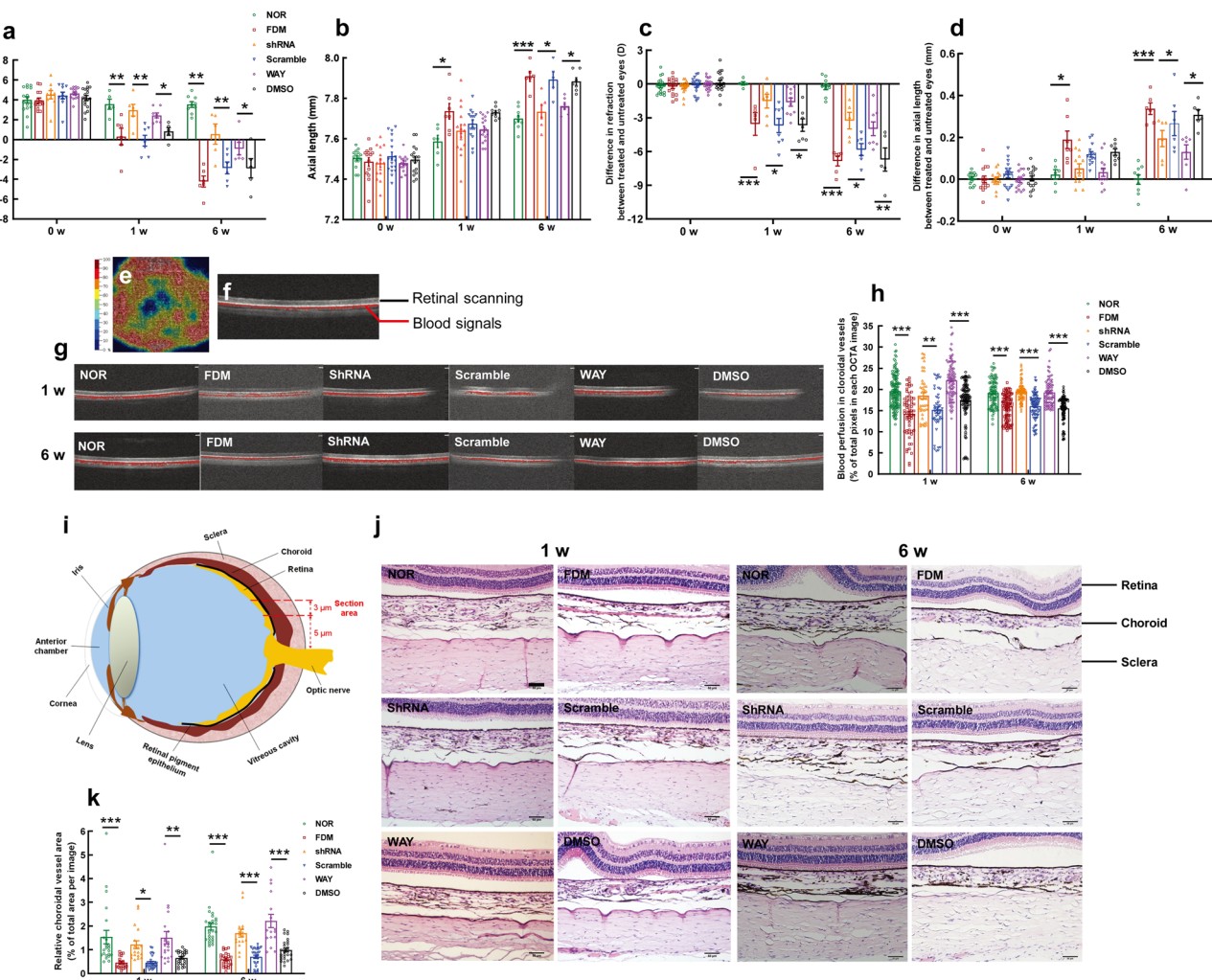

**Fig. 6 Genetic knockdown of *Coch* gene expression or pharmacological blockade of SFRP1 prevented myopia progression by increasing choroidal blood perfusion in the FDM model.** The effects of cochlin shRNA and SFRP1 inhibitor on refraction (**a**) and AL (**b**) in the FDM model at 1 w and 6 w post induction (using data from treated eyes) were shown and compared using two-way ANOVA ($n = 5$–18). The effects of cochlin shRNA and SFRP1 inhibitor on refraction (**c**) and AL (**d**) in the FDM model at 1 w and 6 w post induction (using data of differences between two eyes) were also shown and compared using two-way ANOVA ($n = 5$–18). An en face illustration of OCTA on guinea pig choroid (**e**). A cross-sectional illustration of OCTA of the guinea pig choroid (**f**). Representative OCTA images of the guinea pig choroid at 1 w and 6 w following FDM induction (**g**). Choroidal blood perfusion was quantified from OCTA images (**h**) and compared using one-way ANOVA ($n = 46$–133). A schematic reference to the position of the examined posterior pole tissues (**i**). Representative HE staining images of ocular posterior pole tissues, including choroidal vasculature, at 1 w and 6 w following FDM induction (**j**). Scale bar = 50 μm. The area of the choroidal vessel lumen was quantified from the HE staining images (**k**) and compared by one-way ANOVA ($n = 16$–25). All data represent the mean ± SEM. *$P < 0.05$, **$P < 0.01$, ***$P < 0.001$. SFRP1 secreted frizzled-related protein 1, FDM form-deprived myopia, AL axial length, w week(s), OCTA optical coherence tomography angiography, HE hematoxylin and eosin.

perfusion and diminished choroidal vessel lumen area in the FDM model (Fig. 6e–j), and eventually to the occurrence of nonpathologic myopia. These findings are illustrated in Fig. 8. The axis consisting of different signaling molecules, including cochlin, SFRP1, and CaMKII, and the corresponding tissue cell types, such as retinal photoreceptors, RPE cells, and choroidal vascular endothelial cells, was newly identified and proven to be responsible for myopia pathogenesis.

There are 2 crucial molecular targets on this axis. One is cochlin, a secretory protein comprised of a signal peptide, a cysteine-rich domain homologous to factor C in invertebrates, and 2 domains homologous to noncollagen structural glycoprotein type A (vWFA)[34]. Cochlin is known to be the major non-collagen ECM in the inner ear[35]. Previous studies have shown that vWFA-like domains regulate ECM homeostasis by binding to collagen fibers or glycoproteins and causing collagen

degradation[36,37], indicating that vWFA domain-containing cochlin may also be involved in ECM homeostasis. Indeed, Khetarpal[38] observed that cochlin and mucopolysaccharide accumulated in the vestibular ECM in the inner ear of deaf patients, with a concomitant reduction in type II collagen and an increase in disordered microfibril structures. Moreover, cochlin gene expression is also detected in the eyes of rodents and humans[16]. The excessive deposition of cochlin and its interaction with mucopolysaccharide in the trabecular meshwork ECM contribute to increased resistance of aqueous humor outflow in both glaucoma patients and a glaucomatous rodent model[18]. In this study, we found that the expression of cochlin was dramatically upregulated in the rodent myopia model as early as 1 w post induction (Fig. 3a–c); moreover, the unique position where it resided, the interface between the retinal photoreceptor and RPE (Fig. 3m), rendered it an ideal candidate for the molecular cue

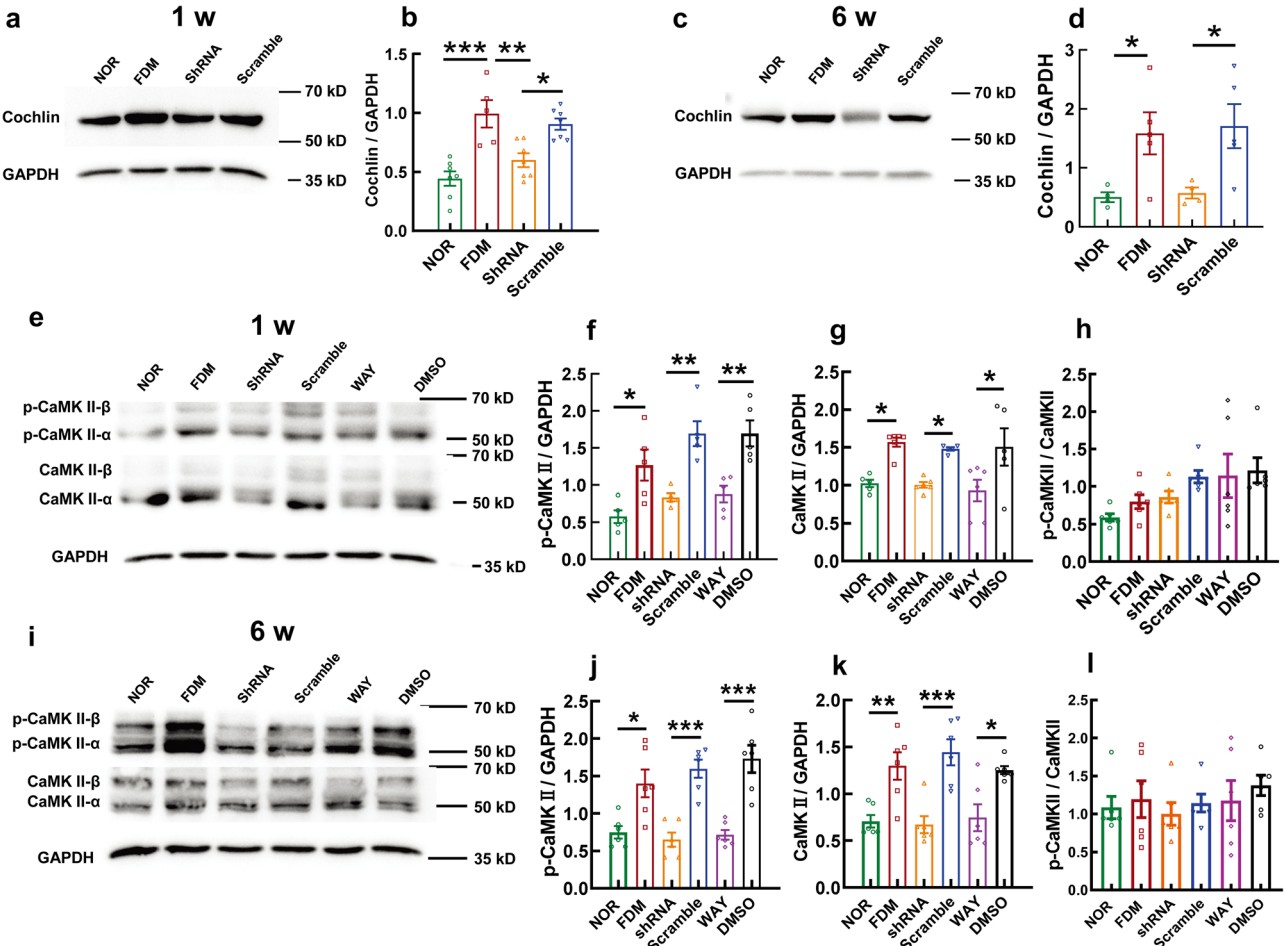

**Fig. 7 Molecular effects of intervening in the two actionable targets, cochlin and SFRP1, in the FDM model.** Representative western blot of cochlin at 1 w (**a**) and 6 w (**c**) following myopic induction. The western blots were quantified in **b** and **d**. The cochlin protein levels were compared using one-way ANOVA ($n = 4$–7). Representative western blot of p-CaMKII and CaMKII at 1 w following myopic induction is shown in **e**. Quantification of the blots (**f**–**h**) was compared using one-way ANOVA ($n = 6$). Representative western blot of p-CaMKII and CaMKII at 6 w following myopic induction is shown in **i**. Quantification of the blots (**j**–**l**) was compared using one-way ANOVA ($n = 6$). All data represent the mean ± SEM. *$P < 0.05$, **$P < 0.01$, ***$P < 0.001$. NOR normal control, FDM form-deprived myopia, CaMKII Ca$^{2+}$/calmodulin-dependent protein kinase II.

that could respond to the blurred image sensed by the photoreceptors, the first-order neurons in retina and transfer the myopigenic information to the RPE, the next relay station. Therefore, we speculate on the basis of literature reports that cochlin, either as a homeooligomer or heterooligomer in ECM[16,39], might interact with transmembrane proteins, such as α-integrin[37] and K$^+$ TREK channel[40], on RPE cells and elicit the overexpression of *SFRP1* through a small GTPase-mediated signaling pathway[41,42]; however, the exact molecular mechanism awaits validation.

The other crucial target on the axis is SFRP1, which has a cysteine-rich domain homologous to the ligand binding site on the Wnt receptors, including Frizzled receptor, low-density lipoprotein receptor-related protein 5 and 6 (LRP5/6), related to tyrosine kinase/Derailed receptor, and retinoid-related orphan receptor[43]. Thus, SFRP1 can directly bind to Wnt proteins and sequester them from binding to their cognate receptors, and is generally deemed an endogenous antagonist of canonical Wnt/β-catenin signaling[44]. However, Courtwright et al.[45] reported that SFRP2, another member of the SFRP family, can directly activate noncanonical Wnt/Ca$^{2+}$ signaling in the endothelial cells of solid tumor vessels. Comparably, in the current study, SFRP-1 elevated the intracellular Ca$^{2+}$ concentration and activated p-CaMKII, the 2 key signaling molecules in the noncanonical Wnt/Ca$^{2+}$

pathway, without eliciting the canonical Wnt/β-catenin pathway (Fig. 5h–o). The reason behind the sole activation of the noncanonical Wnt/Ca$^{2+}$ pathway remains to be determined. Researchers have proposed that the effect of SFRP2 on Wnt signaling may be concentration dependent, with high concentrations leading to Wnt antagonism and low concentrations leading to activation[24]. Hence, it might be that SFRP1 at 30 ng/ml, the concentration measured in the culture media of cochlin-stimulated ARPE-19 cells and human primary RPE cells (Fig. 4i, k), could activate the more sensitive Ca$^{2+}$ signaling but failed to elicit β-catenin signaling. Alternatively, as proposed for SFRP2[24], SFRP1 might activate the Wnt/Ca$^{2+}$ pathway by directly binding to coreceptor LRP5/6 on the vascular endothelial cell membrane. Additionally, neither SFRP-1 nor SFRP-1-containing CM elicited significant changes in CaMKII levels in choroidal vascular endothelial cells (Fig. 5i, n and Supplementary Fig. 3h, m). Therefore, the ratio of p-CaMKII/CaMKII paralleled the expression pattern of p-CaMKII among the experimental groups (Fig. 5m, o), and these results were consistent with previous reports showing activation of noncanonical Wnt signaling in cell cultures[46,47]. In contrast, western blots of posterior pole tissues in the FDM model demonstrated significant variations in CaMKII levels among experimental groups, resembling those of its phosphorylated counterpart (Fig. 7e–g, i–k). This is probably due to

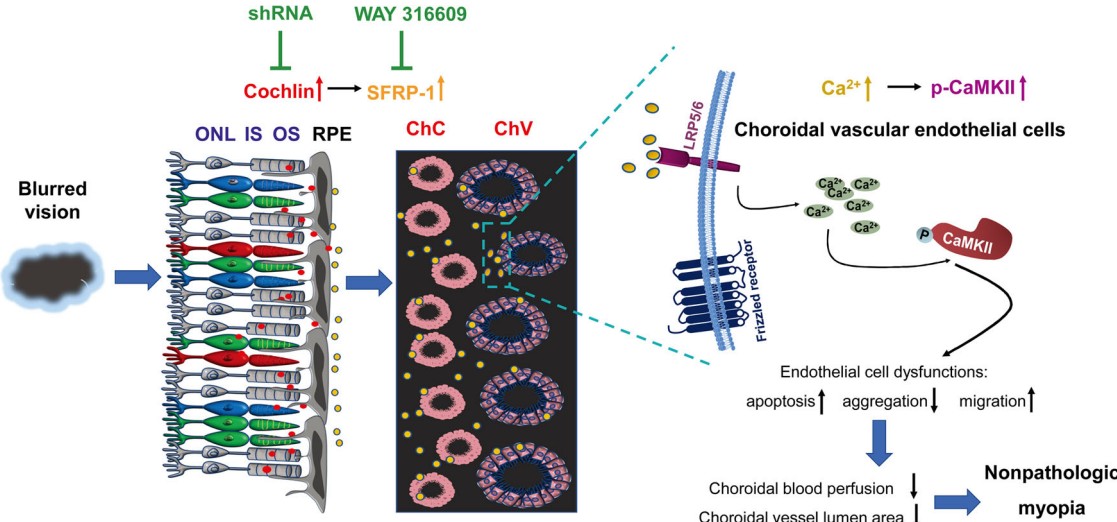

**Fig. 8 Schematic illustration of the cochlin/SFRP1/CaMKII axis identified in this study.** The blurred visual image in the retina induced dramatic upregulation of cochlin (red dots), an ECM protein, at the interface between retinal photoreceptor cells and RPE cells. Cochlin protein stimulated RPE cells to enhance the production and secretion of SFRP1 (orange dots), which in turn acted upon the endothelial cells of ChC and ChV and caused endothelial cell dysfunction, including boosted apoptosis, diminished aggregation, and augmented migration, through activation of the noncanonical Wnt/$Ca^{2+}$/CaMKII signaling pathway. Endothelial cell dysfunctions may elicit reduced vessel lumen area and waned blood perfusion in the choroid, eventually leading to nonpathologic myopia. Targeting cochlin and SFRP1 on the axis precluded the progression of nonpathologic myopia by improving choroidal circulation. RPE retinal pigment epithelium, SFRP1 secreted frizzled-related protein 1, ONL outer nuclear layer, IS inner segments, OS outer segments, ChC choriocapillaris, ChV choroidal vessels, p-CaMKII phosphorylated $Ca^{2+}$/calmodulin-dependent protein kinase II.

tissue heterogeneity of the ocular posterior pole and differential expression profiling of CaMKII in different cell types of the posterior pole.

From a translational perspective, these 2 molecules on the axis could be 2 actionable targets for early intervention of nonpathologic myopia. Cochlin is a large-sized multidomain protein, the crystal structure of which is only partially available[48]. Therefore, the search and design of a specific small molecule to inhibit the activity of cochlin would be infeasible. Thus, it would be prudent to downregulate *COCH* gene expression at the posttranscriptional level. Moreover, the 92% sequence homology between the human *COCH* gene and its guinea pig ortholog supports the preclinical study testing the efficacies of small hairpin RNAs against the human *COCH* gene in guinea pig myopia models. In contrast, SFRP1 is a small protein, and the crystal structure of its family protein is completely available[49]. Therefore, novel small-molecule inhibitors better fitting the activity pocket of SFRP1 and with higher binding affinity and specificity can be designed with computer programs, screened in cell cultures, and tested in animal models.

There are limitations in this study. Due to the difficulty in obtaining ocular posterior pole tissues from subjects with non-pathologic myopia, the expression patterns of cochlin and SFRP-1, the 2 actionable targets identified in this study, have not been validated in clinical samples. Moreover, despite several lines of evidence in this study supporting the role of the cochlin/SFRP-1/CaMKII axis in the pathogenesis of nonpathologic myopia, the possibility that their overexpression may be the consequence of AL elongation cannot be excluded.

In this study, a novel signaling axis constituted by cochlin in retinal photoreceptors, SFRP-1 in RPE, and CaMKII in choroidal vascular endothelial cells was uncovered in FDM and LIM guinea pig models. This axis may shed light on the pathogenesis of nonpathologic myopia. Moreover, targeting cochlin and SFRP1 on this axis prevented myopia progression in the FDM model by improving choroidal circulation, indicating the potential of these molecules as interventional targets for myopia.

## Methods

**Animals.** Four hundred fifty male colored guinea pigs of 3 weeks of age with a body weight of 150–180 g were purchased from Keda Experimental Animal Breeding Center (Wuqing, Tianjin, China) in 3 batches. The guinea pigs were subjected to ocular and optometric examinations upon arrival, and those with eye diseases, myopia, and anisometropia greater than 1.5 D were excluded. The rest were raised at 25 ± 1 °C with a relative humidity of 40-70% in the animal facility and had free access to food, water, vegetables, and vitamin supplements. The illumination was on a 12-h light-dark cycle. All experimental procedures conformed to the Guide for the Care and Use of Laboratory Animals published by the US National Institutes of Health and were approved by the Laboratory Animal Care and Use Committee of Tianjin Medical University (permission number: SYXK 2018-0004).

**FDM and LIM models.** The included guinea pigs were divided into normal control, FDM, and LIM groups ($n = 24$–31/group). The eyes in the normal control group were untreated. Head masks made of opaque latex balloons were generated. The right eye of each guinea pig in the FDM group was fully covered by wearing the head mask, with no further pressure on the eyelid or cornea. The left eye, nose, mouth, and ears of FDM guinea pigs were not covered. During FDM modeling, continuous full coverage and free blinking of the right eyes were ensured.

Each guinea pig in the LIM group wore another type of latex head mask. The eyes, ears, and nose of the animal were exposed, and nylon buckles were sewn onto the head mask around the right eye. Then, a -10.00 D lens was placed on the head mask by nylon buckles in front of the right eye, with the optical center of the lens matching the pupil center. The left eyes of both the FDM and LIM groups were not subjected to myopic induction and served as contralateral controls.

**Ocular biometric parameters**. The ocular biometric parameters, including refraction, AL, and radius of corneal curvature, were measured at appropriate time points following myopic induction.

The pupils of guinea pigs were dilatated with 0.5% compound tropicamide. Refraction was measured by a streak retinoscope (YZ24; Six Six Vision Technology Co., Ltd., Suzhou, Jiangsu Province, China) in the dark with a working distance of 50 cm. The mean refractions of the horizontal and vertical meridian were deemed equivalent spherical degrees. The refraction was measured 3 times for each eye, including the myopia-induced eye (right) and the contralateral control (left), and the average was recorded.

Ophthalmic A-scan ultrasonography (11 MHz, AVISO Echograph Class I-Type Bat; Quantel Medical, Clermont-Ferrand, France) was used to measure AL following topical anesthesia with 0.4% obuvacaine hydrochloride eye drops. Stable wave forms were detected by an ultrasonic probe touching the cornea center perpendicularly. AL was defined as the distance from the anterior cornea apex to the anterior surface of the retina. Each eye was measured 5 times, and the average was recorded.

The corneal curvature radius was measured by a keratometer (OM-4; Topcon, Tokyo, Japan)[50]. A + 8.00 D lens was placed in the front of the keratometer for amplification. The measuring window was parallel to the eye. The keratometer handle and horizontal and vertical knobs were carefully adjusted until the three light circles were in focus and coincided on the cornea. The corneal curvature radius was calculated by averaging the horizontal and vertical readings and then multiplying by 0.451. The average of three measurements was recorded for each eye.

**Sample collection**. At appropriate time points following induction, the guinea pigs were euthanized with an overdose of pentobarbital sodium (100 mg/kg). The eyeballs of some animals were processed for morphological analysis. For gene expression analyses at the transcript and protein levels, the eyeballs were cut at the limbi by a sharp eyebrow trimmer treated by RNase Zap (Thermo Fisher Scientific, Waltham, MA, USA), and the posterior pole tissues were snap frozen in liquid nitrogen and stored at -80 °C.

**Hematoxylin and eosin staining**. The guinea pig eyeballs ($n = 5$-6/group) were fixed in 4% paraformaldehyde, paraffin embedded, and serially sectioned (3 μm thickness/section) at a position starting from 5 μm above the optic nerve. Ten sections were cut on each eyeball, and the sections were stained with HE.

The images were captured under identical optical parameters using the bright field of a BX51 microscope (Olympus Optical Co., Ltd., Tokyo, Japan). The thicknesses of the choroid and sclera in each image were measured using a cellSens Standard electronic system (Olympus Optical Co., Ltd., Tokyo, Japan). Briefly, 3 different positions (left 1/3, middle, right 1/3) on the choroid or sclera were selected in each image, and the thickness at each position was measured by the software. The averaged thickness of the 3 positions represented the thickness of the choroid or sclera on that image. For the analysis of choroidal blood perfusion, the total lumen area of choroidal capillaries and vessels in each image was quantified using ImageJ software (National Institute of Health, Bethesda, MD, USA) and normalized to the total area of the image.

**Total protein extraction**. The posterior pole tissues of eye balls were ground into powder in liquid nitrogen. Then, the tissue powder was lysed with 200 μl of neutral RIPA buffer (50 mM Tris pH 7.4, 150 mM NaCl, 1% NP-40, 0.5% sodium deoxycholate, 0.1% SDS, supplemented with sodium orthovanadate, sodium fluoride, EDTA, and leupeptin, CWBIO, Beijing, China) supplemented with protease and phosphatase inhibitors (CWBIO, Beijing, China). The lysate was subjected to ultrasonication for 1 min, and the resultant solution was centrifuged at 12,000 rpm/min at 4 °C for 20 min. The supernatant was aliquoted, one aliquot was for measuring the total protein concentration, and the others were stored at -80 °C for further usage.

**Measurement of total protein concentration**. The total protein concentration was measured using a BCA protein quantitative kit (CWBIO, Beijing, China). Bovine serum albumin at 2 μg/μl was serially diluted with neutral RIPA buffer (CWBIO, Beijing, China) to generate a standard curve and served as the positive control. RIPA buffer served as the negative control. The extracted protein samples underwent 5-fold dilution with neutral RIPA buffer. Then, 25 μl standards or the diluted protein samples were added to a 96-well plate by duplication, after which 200 μl reaction buffer, composed of liquid A and B at a volume ratio of 50:1, was added to each well. The plate was covered and incubated at 37 °C for 30 min. The absorbance at 562 nm was measured. The protein concentration (μg/μl) was read off the standard curve and multiplied by 5 (the dilution factor).

**Proteomic analysis of the guinea pig ocular posterior pole**. Two hundred fifty micrograms of total protein from each sample were mixed with 300 μl of urea solution (8 M, dissolved in 0.1 M Tris/HCl, pH 8.5, Sigma−Aldrich, St. Louis, MO, USA). The mixture was added to a 10 kD ultrafiltration centrifuge tube and centrifuged at $14,000 \times g$ for 15 min. The ultracentrifugation was repeated 4 times. Then, 45 μl trypsin-containing triethylammonium solution (0.5 M, dissolved in bicarbonate buffer, Sigma−Aldrich, St. Louis, MO, USA) was added to the ultrafiltration tube and incubated with the proteins at 37 °C for 12 h. Finally, the incubation mixture was centrifuged at $14,000 \times g$ for 10 min to collect the alkylated and enzyme-digested proteins. The collected proteins were labeled using an 8-plex iTRAQ kit (SCIEX, Framingham, MA, USA) according to the manufacturer's instructions.

The iTRAQ-labeled polypeptides were redissolved to 100 μl using flow phase A (20 mM $NH_4COOH$, 2 M NaOH, pH 10.0) and loaded onto a chromatographic column (Gemini NX 3U C18 110 A) for separation. The components were collected beginning at a linear gradient within 5-40% of flow phase B (80% acetonitrile solution containing 20 mM $NH_4COOH$ and 2 M NaOH). The linear gradient spanned 90 min. Under the illumination of UV light at 214/280 nm, the components were collected at 1 tube/min, with a flow rate of 200 μl/min. According to the peak type and time period, 24 components were collected, acidified, lyophilized, and analyzed by reverse-phase liquid chromatography coupled with tandem mass spectrometry (LC-MS/MS).

The peptide fragments were redissolved in a solution containing 0.1% formic acid and 2% acetonitrile, mixed, and centrifuged at 13,500 rpm at 4 °C for 20 min. The supernatant was transferred to a loading tube for subsequent identification through mass spectrometry. The peptide fragments were separated at a flow rate of 300 nl/min by a linear gradient formed by flow phase A (5% acetonitrile, 0.1% formic acid) and flow phase B (95% acetonitrile, 0.1% formic acid). Each component was analyzed for 65 min. The isolated peptides were identified by a Thermo Scientific Q Exactive LC-MS/MS system (Thermo Fisher Scientific, Waltham, MA, USA).

ProteinPilot Software 5.0 (SCIEX, Framingham, MA, USA) was used for protein identification and relative quantification. Proteins with a protein reliability greater than 95% and a

coefficient of variation less than 50% were defined as intergroup differentially expressed proteins, which were then analyzed by enrichment of functions (GO analysis) and signaling pathways (KEGG analysis).

**Immunohistochemistry**. The collected and fixed guinea pig eyeballs ($n$ = 5-6/group) were paraffin embedded and serially sectioned (3 μm/section). Ten sections at a comparable position of each eyeball were prepared for IHC. Briefly, following deparaffinization, rehydration, and heat-mediated antigen retrieval, the sections were incubated with a rabbit anti-cochlin primary antibody (Supplementary Table 3) at 4 °C overnight. Then, the sections were washed and incubated with a goat anti-rabbit horseradish peroxidase-conjugated secondary antibody (Supplementary Table 3) for 2 h at room temperature (RT). The sections were counterstained with hematoxylin, mounted with VectaMount (Vector Laboratories, Inc., Burlingame, CA, USA), and observed under a BX51 light microscope (Olympus Optical Co., Ltd., Tokyo, Japan). The pictures were taken using the CellSens Standard electronic system (Olympus Optical Co., Ltd., Tokyo, Japan) with identical optical parameters. On each picture, the intensity of cochlin-positive staining (brown signals) was quantified using ImageJ software (National Institute of Health, Bethesda, MD, USA) and normalized to the area of the retina.

**Immunofluorescence**. The paraffin sections were processed as described above, and incubated with mouse monoclonal anti-rhodopsin and rabbit polyclonal anti-cochlin primary antibodies at 4 °C overnight. After extensive washes, Alexa 647-conjugated goat anti-mouse and Alexa 488-conjugated goat anti-rabbit secondary antibodies were incubated with the sections for 2 h at RT. The sections were then extensively washed and mounted with ProLong anti-fade mounting media with 4′,6-diamidino-2-phenylindole (Thermo Fisher Scientific, Waltham, MA, USA). The images were taken under a Zeiss LSM800 confocal microscope (Zeiss Laboratories, Thornwood, NY, USA). The fluorescence intensity of the green fluorescence was quantified using ImageJ (National Institute of Health, Bethesda, MD, USA) and normalized to the area of the retina.

**Western blots**. The expression of target proteins, including cochlin, p-CaMKII, CaMKII, p-β-catenin, and β-catenin, was examined by western blots ($n$ = 4–16/group). Briefly, 50 μg of total protein from each sample and a multicolor prestained protein ladder (Cat# WJ103, EpiZyme, Shanghai, China) were resolved in a 10% sodium dodecyl sulfate polyacrylamide gel and transferred to a polyvinylidene difluoride membrane. The blots were washed, blocked with 5% nonfat dry milk or BSA, and probed with the corresponding primary antibodies (Supplementary Table 3) at 4 °C overnight. On the next day, the blots were washed and incubated with an appropriate secondary antibody (Supplementary Table 3) at RT for 2 h. The protein signals were visualized with enhanced chemiluminescence plus reagents (Amersham Biosciences, Piscataway, NJ, USA) and imaged using a Multispectral Imaging System (UVP, LLC, Upland, CA, USA). The optical densities of cochlin, p-CaMKII, CaMKII, p-β-catenin, and β-catenin were quantified using ImageJ (National Institute of Health, Bethesda, MD, USA) and normalized to that of an internal standard, glyceraldehyde 3-phosphate dehydrogenase (GAPDH). The ratios of p-CaMKII/CaMKII and p-β-catenin/β-catenin were also calculated.

**Extraction of total RNA and reverse transcription**. The guinea pig ocular posterior pole tissues, 2 RNase-free zirconia beads with a diameter of 4 mm and 3 beads with a diameter of 3 mm

(Wuhan Servicebio Technology Co., Ltd., Wuhan, Hubei Province, China), and the appropriate amount of lysis buffer from the Tissue RNA Extraction Kit (EZMED, Suzhou, Jiangsu Province, China) were placed in a 2-ml hardened microcentrifuge tube, which was then securely positioned into a prechilled KZ-II high-speed tissue grinder (Wuhan Kangtao Technology Co., Ltd., Wuhan, Hubei Province, China). The tissue was completely ground for 5 min under properly set parameters, and total RNA was extracted from tissue homogenates following the manufacturer's instructions. Total RNA from human ARPE-19 cells was extracted using a GeneJET RNA Extraction Kit (Thermo Fisher Scientific, Waltham, MA, USA). The concentration and purity of total RNA were examined using a Nanodrop 2000 (Thermo Fisher Scientific, Waltham, MA, USA).

After thorough digestion with DNase I, 1 μg of total RNA was reverse transcribed using random hexamer primers and reagents in a RevertAid cDNA synthesis Kit (Thermo Fisher Scientific, Waltham, MA, USA) according to the manufacturer's protocol.

**Quantitative real-time PCR**. The expression of the guinea pig *Coch* gene in ocular posterior pole tissues and the *SFRP1* gene in human RPE cells was examined using qPCR in an HT7900 Real-Time PCR System (Applied Biosystem, Foster City, CA, USA). The cDNA content of the *Coch* gene was normalized to an internal standard β-actin gene, whereas that of the *SFRP1* gene was normalized to another internal standard *GAPDH* gene. The 25-μl reaction mixture was composed of 4 μl cDNA template, gene-specific qPCR primers (Supplementary Table 4), and 12.5 μl EvaGreen 2X Master Mix (ABM, Richmond, BC, Canada). Two microliters of cDNA from each sample were pooled, serially diluted, and used as templates to generate a standard curve between Ct values of the examined gene and logarithms of cDNA template concentrations. The standard curves demonstrated a similar priming efficiency between the gene to be examined and the corresponding internal standard gene. The standard curves also served as positive controls for qPCR, and the reactions using water as template served as negative controls. The qPCR program consisted of preincubation at 50 °C for 2 min, denaturation at 95 °C for 10 min, 40 cycles of denaturation at 95 °C for 15 s and extension at 60 °C for 1 min. A dissociation stage was added to check amplicon specificity. The relative expression levels of the examined genes were analyzed using a comparative threshold cycle ($2^{-\Delta\Delta Ct}$) method.

**Cell cultures and treatments**. The human RPE cell line ARPE-19 and the simian choroidal vascular endothelial cell line RF/6 A were purchased from the Chinese Academy of Science (Shanghai, China) and maintained in complete culture media consisting of Dulbecco's modified Eagle's medium (DMEM)/F-12 (Thermo Fisher Scientific, Waltham, MA, USA), 10% fetal bovine serum (Thermo Fisher Scientific, Waltham, MA, USA), 100 U/ml penicillin, 100 μg/ml streptomycin (Thermo Fisher Scientific, Waltham, MA, USA), and 2 mM L-glutamine (Thermo Fisher Scientific, Waltham, MA, USA) at 37 °C and 5% $CO_2$ in a humidified incubator. Cells at passages 3-5 following resuscitation were used for the experiments.

Primary human RPE cells were purchased from Hunan Fenghui Biotechnology Co. Ltd. (Changsha, Hunan Province, China). The purity of the primary RPE cells was tested by immunofluorescence staining with an antibody against CK-8, a RPE cell-specific marker, as well as by STR analysis to ensure that there was no contamination of other cell types or cell lines. Primary RPE cells at passages 2–3 were used for the experiments.

ARPE-19 cells were seeded into a 12-well plate (Corning Costar, Cambridge, MA, USA) at a density of $4 \times 10^5$ cells/ml and

were divided into normal control and cochlin-treated groups ($n = 6$/group). The next day, the cells in each well reached 90% confluency. The complete culture media containing recombinant human cochlin protein (10 µg/ml, R&D Systems, Minneapolis, MN, USA) were passed through a 1-ml syringe with a 30-gauge needle 20-30 times, allowing for oligomerization of the recombinant protein[19,41], after which the ARPE-19 cells were incubated with cochlin oligomer-containing culture media; the normal controls were incubated with complete culture media. Twenty-four hours later, the 2 groups of cells were collected. One part ($n = 3$/group) was processed for microarray analysis, and the other was processed for qPCR validation ($n = 3$-6/group). The culture media were collected for measuring the target protein concentration using an ELISA or for the CM to stimulate RF/6 A cells in the following functional assays. Primary human RPE cells were also employed to validate gene expression via qPCR and ELISA.

**Microarray analysis.** The total RNA of the collected RPE cells was extracted and reverse transcribed as described above. Then, the synthesis of antisense RNA (aRNA) was driven by the T7 promoter, during which amino-allyl-UTP was added to form amino-allyl-aRNA (aa-aRNA). NHS-CyeDye was subsequently added to convert aa-aRNA into CyeDye-aRNA, whereby in vitro transcription and labeling were conducted. After purification, the dye-labeled aRNA was hybridized with Phalanx OneArray. The hybridized array was cleaned, scanned, and analyzed.

The Rosetta Resolver System (Rosetta Commons, USA) was used to perform preprocessing (including data filtering and correction) and calculation of the data from the Gene Expression Profile Detection Chip. In the environment of R 3.0.3, amap, ctc, gplots, and other components were used for cluster analysis, and the built-in "prcomp function" of R was used for principal component analysis. The screening criterion for DEGs was $\log_2 |$ fold change $| > 1$.

**ELISA.** The protein levels of SFRP1 in the culture media of ARPE-19 or human primary RPE cells treated with or without recombinant human cochlin ($n = 5$-6/group) were measured using an ELISA kit (Biorbyt, Cambridge, UK, Cat# orb442167). Briefly, an antibody specific to SFRP1 was precoated on a 48-well microplate. After SFRP1 in the samples bound to the precoated antibody, a biotin-conjugated antibody specific to SFRP1 and horseradish peroxidase-conjugated avidin were sequentially added to sandwich the bound SFRP1. Then, 3,3′,5,5′-tetra-methylbenzidine substrate was added for quantification. The standard curve was generated by serial dilutions of recombinant SFRP1 and served as positive controls, whereas the wells with diluent only served as negative controls. The optical density was measured at 450 nm by an Infinite 200 PRO Multimode Micro-plate Reader (Tecan Group Ltd., Männedorf, Switzerland). The concentration of SFRP1 (ng/ml) was calculated according to the standard curve.

**Grouping of choroidal vascular endothelial cells for functional assays.** To examine the direct effects of SFRP-1, the RF/6 A cells used for functional assays were divided into 6 groups ($n = 6$/group): (1) Normal group: RF/6 A cells were cultured with complete culture media; (2) SFRP-1 group: RF/6 A cells were incubated with 30 ng/ml SFRP1 (human recombinant protein, stock solution 250 µg/ml in PBS, R&D Systems, Minneapolis, MN, USA) in complete culture media; (3) SFRP-1 + WAY 316606 (SFRP-1 + WAY) group: RF/6 A cells were incubated with 30 ng/ml SFRP1 and 2 µM WAY 316606 (stock solution 10 mM in DMSO, MCE, Shanghai, China) in complete culture

media. WAY 316606 acts as a specific inhibitor of SFRP1 at this concentration in cell cultures[51,52]; (4) SFRP-1 + DMSO (SFRP-1 + DMSO) group: RF/6 A cells were incubated with 30 ng/ml SFRP1 and 0.02% DMSO (Sigma–Aldrich, St. Louis, MO, USA) in complete culture media; (5) SFRP-1 + Wnt3a (SFRP-1 + Wnt3a) group: RF/6 A cells were incubated with 30 ng/ml SFRP1 and 0.3 µg/ml Wnt3a (human recombinant protein, stock solution 200 µg/ml in PBS, R&D Systems, Minneapolis, MN, USA) in complete culture media. Wnt3a is an activator of the canonical Wnt signaling pathway[27]. (6) SFRP-1 + PBS (SFRP-1 + PBS) group: RF/6 A cells were incubated with 30 ng/ml SFRP1 and 0.15% sterile PBS (Thermo Fisher Scientific, Waltham, MA, USA) in complete culture media.

To examine the effects of CM from cochlin-stimulated ARPE-19 cells, RF/6 A cells were divided into 6 groups ($n = 6$/group): (1) CM group: RF/6 A cells were cultured with complete culture media; (2) Cochlin CM group: RF/6 A cells were cultured with CM from ARPE-19 cells stimulated with cochlin; (3) Cochlin CM + WAY 316606 (Cochlin CM + WAY) group: RF/6 A cells were cultured with CM and 2 µM WAY 316606; (4) Cochlin CM + DMSO (Cochlin CM + DMSO) group: RF/6 A cells were cultured with CM and 0.02% DMSO; (5) Cochlin CM + Wnt3a (Cochlin CM + Wnt3a) group: RF/6 A cells were cultured with CM and 0.3 µg/ml Wnt3a; (6) Cochlin CM + PBS (Cochlin CM + PBS) group: RF/6 A cells were cultured with CM and 0.15% sterile PBS.

**Proliferation assay by Cell Counting Kit-8.** RF/6 A cells were seeded into a 96-well plate (Corning Costar, Cambridge, MA, USA) at a density of $8 \times 10^3$ cells/well and were grouped and treated as described above for 24 h. Then, cell proliferation was examined using a CCK-8 (Dojindo Laboratories, Kumamoto, Japan), which demonstrated equal accuracy and reliability in quantifying cell proliferation to the Edu assay[53], one of the gold standards for analyzing cell proliferation. Briefly, 100 µl of complete culture media containing 10% CCK-8 replaced the original media in each well and was incubated with the cells under 5% $CO_2$ at 37 °C for 3 h. Then, the absorbance at 450 nm ($OD_{450}$) was measured by an Infinite 200 PRO Multimode Microplate Reader (Tecan Group Ltd., Männedorf, Switzerland). Wells containing complete culture media and CCK-8 but no cells served as empty controls. Cell proliferation was presented as the percentage of the normal or CM group.

**Caspase 3/7 activity assay.** RF/6 A cells were seeded into a black 96-well plate (Corning Costar, Cambridge, MA, USA) at a density of $1 \times 10^4$ cells/well and were grouped and treated as described above for 24 h. Then the cells in each well were lysed with 100 µl of neutral RIPA buffer on ice for 20 min in the dark. After thorough mixing, 10 µl of cell lysate was removed and diluted with 15 µl of neutral RIPA buffer, and subjected to BCA total protein quantification as described above. The caspase 3/7 activity in the remaining cell lysate was measured by an Apo-ONE Homogeneous Caspase 3/7 Assay (Promega, Madison, WI, USA). Ninety microliters of assay reagent were mixed with an equal volume of cell lysate, and the mixture was incubated at RT for 4 h. Wells containing only RIPA buffer and Apo-ONE assay reagent but no cells served as empty controls. The fluorescence intensity was measured by an Infinite 200 PRO Multimode Microplate Reader (excitation 499 nm, emission 521 nm, Tecan Group Ltd., Männedorf, Switzerland). The fluorescence intensity in each well was subtracted by the average of empty controls and normalized to the total protein concentration (µg/µl).

**Annexin V apoptosis assay**. RF/6 A cells were seeded into 24-well plates (Corning Costar, Cambridge, MA, USA) at a density of $1 \times 10^5$ cells/well and were grouped and treated as described above. Twenty-four hours later, the cells were stained with FITC-conjugated annexin V (FITC Annexin V Apoptosis Kit, Thermo Fisher Scientific, Waltham, MA, USA) according to the manufacturer's protocol. The cells were fixed with 1% paraformaldehyde (Sigma−Aldrich, St. Louis, MO, USA). The percentage of cells positive for annexin V staining was detected with a BD FACSCelesta Cell Analyzer (BD Biosciences, San Jose, CA, USA) and analyzed by FlowJo software (Becton, Dickinson and Company, Ashland, OR, USA)[54].

**Matrigel assay**. RF/6 A cells were seeded in 6-well plates at a density of $8 \times 10^5$ cells/well and were grouped and treated as described above for 24 h. The Matrigel matrix basement membrane (Corning Costar, Cambridge, MA, USA) was thawed at 4 °C overnight and transferred to an ice-cold 24-well plate with 300 μl/well using prechilled pipette tips. The Matrigel was solidified at 37 °C for 30 min and covered with 500 μl DMEM/F12 culture media. Then, the covering media were removed, and the cells were seeded in each gel well by transferring $5 \times 10^4$ cells in 500 μl corresponding media from the 6-well plates. The cells were incubated for 8 h, and pictures were taken using the CellSens Standard electronic system (Olympus Optical Co., Ltd., Tokyo, Japan) under a light microscope (BX51, Olympus Optical Co., Ltd., Tokyo, Japan) with identical optical parameters. Each well was divided into 4 quadrants, and 3 representative pictures were acquired for each quadrant. The numbers of cell aggregates in the 3 pictures were averaged to represent the aggregate number in each quadrant; then, the aggregate numbers in all quadrants were summed to represent the total number of cell aggregates in each well.

**Transwell assay**. The RF6/A cells were cultured in 6-well plates at a density of $8 \times 10^5$ cells/well and were grouped and treated as described above for 24 h. Afterwards, a Transwell assay was performed using Transwell chambers (pore size 8.0 μm, diameter 6.5 mm) in a 24-well plate (Corning Costar, Cambridge, MA, USA). The upper chambers were balanced with 100 μl DMEM/F12 for 1 h at 37 °C. Then, each chamber was seeded with $1 \times 10^5$ cells in 100 μl corresponding media from the 6-well plates, whereas each well of the bottom plate was filled with 600 μl complete culture media. The Transwell system was incubated for 8 h as described above. Then, the cells were rinsed with PBS, fixed and stained with 0.1% crystal violet (dissolved in anhydrous ethanol, Sigma−Aldrich, St. Louis, MO, USA) for 20 min and washed 3 times with tap water. The cells at the inner bottom of the chamber were wiped off with a dampened cotton swab. The chambers were air dried, and the bottom membranes were circumcised and mounted on slides with neutral balsam (Sigma−Aldrich, St. Louis, MO, USA). Pictures were taken, and the number of migrated cells on each membrane was quantified as described above.

**Intracellular Ca²⁺ concentration**. RF/6 A cells were seeded into a black 96-well plate (Corning Costar, Cambridge, MA, USA) at a density of $4 \times 10^4$ cells/well, and normal, SFRP-1, SFRP-1 + WAY 316606, and SFRP-1 + DMSO groups of cells were included and treated as described above. At 24 h post treatment, the culture media were removed, and the CCK-8 assay was conducted within the next hour. Based on the results of the proliferation assay, 24 h of SFRP-1 treatment did not affect cell proliferation; hence, the OD 450 value could serve as a representative of cell number for normalization. Then, the intracellular Ca²⁺ concentration was measured according to the protocol of the Fluo-8 Medium Removal Calcium Assay Kit (Abcam, Cambridge, MA, USA). Briefly, 100 μl Fluo-8 dye-loading solution was added to each well after removal of culture media, and the covered 96-well plate was then incubated at 37 °C for 30 min followed by another 30 min incubation at RT. The dye-loading solution was replaced by DMEM/F-12 media. After further incubation at 37 °C for 5 min, the fluorescence intensity (excitation 490 nm, emission 525 nm) was measured by an Infinite 200 PRO Multimode Microplate Reader (Tecan Group Ltd., Männedorf, Switzerland) and normalized to the OD 450 value obtained from the CCK-8 assay. The relative intracellular Ca²⁺ concentration was expressed as a percentage of the normal control.

**Preparation of lentiviruses overexpressing the guinea pig *Coch* gene and expressing shRNAs against its transcript**. The cDNA of the guinea pig *Coch* gene (NM_001173043.1), 3 shRNAs against the transcript of this gene, and a scramble shRNA were inserted into lentiviral expression vectors. The expression of guinea pig *Coch* cDNA was driven by a cytomegalovirus promoter, whereas the expression of each shRNA was driven by a U6 promoter. The sequences of the shRNAs are listed in Supplementary Table 4. After construction, the lentiviruses carrying *Coch* cDNA and the shRNAs were packaged, concentrated, and titered by Beijing Mijia Technology Co., Ltd. (Beijing, China).

**Selection of the most efficient shRNA for the *Coch* transcript**. The 293 T cells were seeded in 6-well plates at $5 \times 10^5$ cells/well and divided into 4 groups, including shRNA1 + *Coch* overexpression, shRNA2 + *Coch* overexpression, shRNA3 + *Coch* overexpression, and Scr + *Coch* overexpression groups ($n = 6$/group). Each well of cells was transduced with the corresponding lentiviruses at $3 \times 10^6$ TU/ml/virus with the assistance of polybrene (Sigma−Aldrich, St. Louis, MO, USA). At 64 h post transduction, the cells were collected for RNA extraction, reverse transcription, and qPCR analysis of guinea pig *Coch* gene expression as described above. The comparable fluorescence intensity of green fluorescence protein following lentiviral infection suggested the similar transduction efficiency of the 293 T cells with the lentiviruses (Supplementary Fig. 7a). The qPCR results showed that guinea pig *Coch* transcript levels in the shRNA1+ COCH group were significantly lower than those in the shRNA2 + COCH group and lower, albeit not significantly, than those in the shRNA3 + COCH group. (All $P < 0.001$, shRNA1 + COCH vs Scr + COCH, shRNA2 + COCH vs Scr + COCH, shRNA3 + COCH vs Scr + COCH; both $P < 0.05$, shRNA1 + COCH vs shRNA2 + COCH, shRNA3 + COCH vs shRNA2 + COCH; Supplementary Fig. 7b). Therefore, shRNA1 could most efficiently knockdown guinea pig *Coch* gene expression, and was selected for the following in vivo experiments.

**Grouping and treatments of animals for examining the effects of targeting the cochlin/SFRP1/CaMKII axis in the FDM model**. The guinea pigs were divided into 6 groups: the normal control, FDM, shRNA + FDM, Scr + FDM, WAY + FDM, and DMSO + FDM groups. The normal control group received no treatment. The FDM group was subjected to form deprivation as described above. The shRNA + FDM and Scr + FDM groups were intravitreally injected into the right eyes at 3 d prior to FDM induction with 3.5 μl lentiviral particles carrying shRNA1 and Scr, respectively, with the viral titers equally adjusted to $1 \times 10^8$ TU/ml. Intravitreal injection of the cell culture media that had been used to dilute the lentiviral particles was performed at 3 w following myopia induction. The WAY + FDM and DMSO + FDM groups were intravitreally injected with 3.5 μl WAY 316606

(2 μM) and DMSO (0.02%), respectively, in the right eyes, and the injections were performed 3 d before and 3 w after FDM induction. At 1 and 6 w following induction, ocular biometric parameters, including refraction ($n = 5$–18/group) and AL ($n = 5$–18/group), were measured, and OCTA was performed ($n = 45$-135 images/group). Then, the eyeballs and posterior pole tissues were collected for choroidal vessel morphology ($n = 20$-30 images/group) and western blot analyses ($n = 4$-8/group), respectively, as described above.

**Intravitreal injections in the FDM animals.** Intravitreal injections were performed in the surgery room of the animal facility. The guinea pigs were anesthetized with an intraperitoneal injection of pentobarbital sodium (40 mg/kg), their eyes were locally anesthetized with oxybuprocaine hydrochloride, and pupils were dilated with tropicamide. The guinea pigs were then injected in the right eyes at 2 mm posterior to the limbus by a 30-gauge needle mounted on a Hamilton syringe (Sigma–Aldrich, St. Louis, MO, USA). The needle was kept in the vitreous cavity for 15 s and gently withdrawn to avoid leakage. Ofloxacin drops (Akorn, Inc. Lake Forest, IL, USA) were topically administered to prevent bacterial infection.

**OCTA analysis of the FDM animals.** The guinea pigs were anesthetized by an intraperitoneal injection of pentobarbital sodium (30 mg/kg), fully mydriatic with tropicamide, and secured in a prone position on a small mounting table. The right eyes were exposed, and the corneas were moistened to generate a smooth optical interface. The OCTA examination was conducted by aligning the detection probe of RTVue XR Avanti (Optovue company, Fremont, CA, USA) to the eyeball and adjusting the focal length to clearly present the view of the fundus. Then, the Angio retina 3 × 3 mode in AngioVue software was employed to perform 304 B-scans at equal intervals along the x- and y-axes, with each B-scan containing 304 A-scans. The choroidal blood flow of the guinea pig was measured, and 5 images were captured for each eye.

The OCTA images were analyzed using Python (version 3.7.9, Python Software Foundation, Beaverton, OR, USA). Briefly, the pixel matrix of an OCTA image was extracted, and the number of pixels representing blood flow was calculated based on the RGB difference. Then, the image was converted to grayscale mode and denoised using the Gaussian noise reduction algorithm, and the number of pixels in the choroidal region was calculated according to the grayscale difference. Finally, the pixel number of blood flow was normalized to that of the choroid region.

**Statistics and reproducibility.** Statistical analyses were performed using Statistical Program for Social Sciences 20.0 (IBM SPSS Inc., New York, NY, USA). The data were examined by D'Agostino and Pearson omnibus normality tests to confirm a Gaussian distribution, after which they were examined by Levene's test to confirm homogeneity of variance. All data are expressed as the mean ± SEM. Then, the differences between the experimental groups were analyzed by one-way or two-way ANOVA followed by Tukey's post hoc test. A P value less than 0.05 was considered statistically significant.

**Reporting summary.** Further information on research design is available in the Nature Portfolio Reporting Summary linked to this article.

## Data availability

The original proteomic data have been uploaded to the iPoX website and released to the public with an ID of IPX0005328000. Since the iProX is an official member of ProteomeXchange Consortium, the proteomic dataset also has a ProteomeXchange identifier: PXD037872. The original microarray data have been uploaded to the ArrayExpress website with an accession ID E-MTAB-12364, and these data have been released to the public and can be viewed via the link https://www.ebi.ac.uk/biostudies/arrayexpress/studies/E-MTAB-12364?key=07f125bd-089e-4809-800e-ac7ae892de92. The source data underlining the main figures presented in the manuscript are provided in Supplementary Data 1.

## Code availability

The codes for analyzing OCTA images are shared in Zenodo (https://zenodo.org/record/8218501).

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

## Acknowledgements

We thank Dr. Xun Liu, Ms. Yuejun Zhou, Ms. Nianen Liu, Mr. Jiaxing Chi, and Ms. Xueyi Pang for technical assistance. This work was supported by the National Natural Science Foundation of China (81970827), Tianjin high-level talent selection and training project in the health industry (TJSJMYXYC-D2-042), Tianjin Medical University Eye Hospital High-level Innovative Talent Program (YDYYRCXM-B2023-01), and the Tianjin Key Medical Discipline (Specialty) Construction Project (TJYXZDXK-037A).

## Author contributions

Y.Z. conceived, designed, and supervised the project, obtained the funding, wrote versions of the manuscript, and formatted and submitted the manuscript. C.G., S.L., J.W., Sennan W., and W.Z. performed the experiments and acquired the data. H.R. wrote the computer codes and analyzed the OCTA results. Sennan W., W.Z., and S.L. performed the morphometric analysis of the stained sections of choroidal vessels. Y.C., Shuqing W., Y.Z., and Z.L. conducted statistical analyses. Z.L. supervised the use of the OCTA device. All authors reviewed and approved the manuscript.

## Competing interests

The authors declare no competing interests.
