## [Peer Review File · Communications Biology]

Reviewers' comments:

Reviewer #1 (Remarks to the Author):

This is a timely, interesting and important manuscript that reports significant and new findings implicating involvement of Cochlin and SFRP1 towards development of myopia. Manuscript, despite providing strong evidence in support of reported findings/conclusions, necessitates revision for further serious consideration.

1) Abstract does not report lens induced myopia (LIMS) model. Authors should also validate a subset of data using primary RPE cells and abstract need to incorporate the same.

2) Abbreviations should be spelled out at their first appearance throughout the text.

3) Page 23, line 575, 577, What is diluent? The composition of diluent needs to be described.

4) Page 24, line 612, please write make and model of mass spec.

5) Page 24, line 615, ProteinPilot 4.0 is not offered by Thermo. Authors need to correct this information.

6) Page 27, line 693, did they use any other RPE cells? It will be important to use some primary RPE cells (preferably human) to validate at least a subset of experiments.

7) Please review the reference style including font for formatting conformity.

8) Page 44, table 1, MAP1B value is 0.000, either it should be eliminated, or a real value be included.

9) Page 48, Figure 1, Perhaps authors can consider including a cross section of eye cartoon synthesizing measurements 1A-E and pictorially depicting what will myopia look like with respect to measurements in a cross-sectional view? This can be included in current void space in the middle of the figure panels.

10) Page 49, Figure 2B, can they replace IDs with gene names? or include gene names side by side?

11) Page 50, Figure 3 A-C, please consider labeling panel A with a different color so that it is clear to the readership.

12) Page 50, Figure 3, Panel G and throughout for Western blot-can authors please provide the whole blot as supplemental figures?

13) Page 53, Figure 7, Please consider providing a cartoon of eye cross section indicating which location of the retina has been presented in G and H. This can be placed adjacent to empty space next to Figure H.

14) Page 55, Figure 8, The summary cartoon should have some rethink. For example, the blurred vision should perhaps be at the end after the arrow of Non pathologic myopia.

In summary, the manuscript needs revisions, light English editing for brevity and better presentation. Authors should consider using primary RPE for validation and provide Western blots as supplements for review of the readership. The findings of molecular links to myopia are significant advancements which is likely to exert wide impact in ophthalmology and vision science as well our understanding of cell/tissue signaling.

Reviewer #2 (Remarks to the Author):

This manuscript addresses the molecular mechanisms that may underlie myopia. The authors use the guinea pig to model form-deprived myopia (FDM) and lens-induced myopia (LIM) and test the possible contribution of scleral hypoxia as a cause of myopia.

Through proteomic studies the authors identify colchlin, an ECM component, as a protein up-regulated in both models. Using different approaches, they show that the protein localises to the retina and perhaps the RPE. Using a RPE derived cell line, ARPE-19, they show that colchlin addition promotes the up-regulation of the secreted protein SFRP1, which, in turn, stimulates CaMKII in RF/6A, a cell line derived from choroid vasculature. They conclude that a so called "axis" composed of cochlin in the retina, SFRP1 in the RPE, and CaMKII in choroidal vascular endothelial cells is responsible for non-pathologic myopia and suggest that cochlin and SFRP1 are potential therapeutic targets for myopia.

This is a potentially interesting study. However, the overall rationale of the study is difficult to follow and the logic followed by the authors is difficult to grasp especially for non-specialists. The conclusions are largely overstated especially because some of the critical experiments are based on the use of cell lines that are far from representing the in vivo situation. In addition, there are factual mistakes that should be corrected. Overall the authors should re-elaborate their manuscript to facilitate understanding the logic they followed. For example, some of the information provided in the discussion would be better placed in the introduction (e.g. SFRP1 has been associated to myopia in GWAS studies) or to explain why the picked one gene/protein vs any other. The conclusions should be toned down as they study simply show correlations. The up-regulation of the indicated "axis" may simply be a consequence and not a cause to myopia. Additional specific comments are listed below:

1. The introduction is rather focused and does not clearly explain what the rationale of the study.
2. There is little information on how the guinea pig form of deprived myopia (FDM) and lens-induced myopia (LIM) were established. The related methods are copy paste of those described in ref 44 and these in turn come from an additional reference of the authors from 2011. This information is critical to reproduce the study and it should be described better.
3. Line 157. The authors mention that colchlin gene expression is up-regulated in both FDM and LIM. Besides the fact that they have mostly looked at the protein, there is need to explain what the gene encodes for and why they have focused on this protein in particular vs all the others they found up or down regulated in both models. It will be also important to understand how this up-regulation occurs in the LIM model
4. Table 1, the name of the genes and the name of the proteins are mixed up. For example, the name of the colchlin gene is COCH whereas the protein is called colchlin. Similarly, the KRT19 gene encode keratin 19...etc. The table needs to be corrected
5. Line 202. The authors suggest that Cochlin may be "an early interventional target for non-pathologic myopia". At this point of the study this is an overstatement. There is only a correlation between Cochlin over-expression and myopia. No evidence that this is the cause of myopia.
6. It is totally unclear why the authors have decided to stimulate the ARPE-19 (and not APE-19 as indicated in line 207) cells with colchin. What is the rationale behind? This should be explained. The ISH distribution does not support a localisation in the RPE, as indicated by the authors, as only minimal staining is observed in this layer. On the contrary, the protein seems to localise in the outer nuclear layer (among photoreceptor cell bodies) and perhaps in the initial portion of the outer segment. Why not testing the effect on photoreceptors? Furthermore, ARPE-19 cells assume some RPE characteristics only if properly stimulated. It is unclear if this has been done, no information is available in the methods. The authors cannot use ARPE-19 and RPE interchangeably because this is not the case.
7. Again, why did the authors focus on SFRP1 and not in any of the other genes that were up-regulated upon APRE-19 treatment with colchin? Perhaps indicating that SFRP1 has been associated with myopia in GWAS studies will help. Addressing the distribution of SFRP1 will also be relevant here.

8. What is the rationale of incubating RF/6A cells with the condition media of colchin treated APRE-19 cells? There is no explanation.
9. SFRP1 is commercially available. The authors need to treat the cells with SFRP1 to demonstrate the possibility that this protein induces apoptosis of RF/6A cells. All experiments with conditioned media are indirect. Furthermore, why focusing on apoptosis and tubulogenesis?
10. What is the Matrigel assay? This should be explained to allow non-experts to follow the logic of the study.
11. Line 276. Again, the authors overstate the value of their results. The aggregates shown in figure 5D cannot be considered as "tubules" nor what observed in RF/6A cells can be translated to the choroid vasculature as such.
12. Line 287 The authors consider that canonical Wnt signalling was activated in the RF/6A cells upon exposure to Wnt3a. This seems an erroneous conclusion. The phosphorylated form of beta-catenin increases when the pathway is inactive and thus the protein is targeted for degradation.
13. The data supporting CaMKII activation in the presence of the condition medium are unclear. The biological properties of CaMKII are regulated by multi-site phosphorylation. What should be the mechanism? Does addition of SFRP1 alone mimic CaMKII hypothetical activation? How does CaMKII become over-expressed? SFRP1 has been shown to act as a multifunctional modulator of Wnt signalling as well as to regulate other pathways (Notch) as well as to interact with other proteins. This should be mentioned and considered at least in the discussion.

Reviewer #3 (Remarks to the Author):

The authors conducted a comprehensive experiment and try to figure out the role of cochlin/SFRP1/CaMKII axis in FDM/LIM development. At first, proteomic analysis of ocular posterior pole had identified that cochlin was the strongest DEG. Then the time dependent expression patterns of cochlin were validated by a series of experiments and the RPE was found to be expressing this protein; By further stimulating the RPE with cochlin, SFRP1 was found to be highly expressed. The authors also demonstrated how cochlin and SFRP1 influence the choroidal vascular remodeling and their effect on FDM formation. These findings are important to unveil the biological mechanism of myopia development, especially important to figure out the 'RPE and choroid mechanisms' in myopia development.

Overall, this study is well designed, with clear logic, proper experimental techniques applied and sufficient in vivo and in vitro evidence. Meanwhile, the manuscript is well written, the figures are clear and well organized.

Concerns:

Cochlin is a secretory ECM protein and was found in glaucomatous trabecular meshwork, but it's role in myopia development is unclear. Although this study is driven by omic analysis, which is hypothesis free in the initial stage, a clear hypothesis should be form for following analysis. Whether cochlin is a consequence induced by axial elongation stress or it is caused by blur vision remain unclear. More importantly, the mechanism that how cochlin is interplay with SFRP1 and CaMKII and the potential signaling pathway is not specified. It will be good to have a theoretical discussion and mention the limitation of this study in the discussion section.

Minor:

1. In the introduction section, it will be nice to include background knowledge of cochlin and the hypothesis of cochlin and myopia development.
2. In the section of 'Validation of Cochlin expression in retinas of FDM and LIM models': From line 179-185, authors explained the difference of cochlin protein expression levels in the outer segments of the ocular at 6w time points, however, in the coming section(from line 196), authors only demonstrated it's expression pattern of the first 3 weeks. It will be more clear if the results were displayed in a chronological order, e.g the 1) validation of the expression at early stages; 2) late stages; 3) location

of cochlin protein expression;

3. Figure 3M, please shift the the x-axis to the same scale of Figure 3N if they the measurements are taken from the same samples.

4. Line 223: Miss spell 'SPRP1'.

Dear reviewers:

Thank you very much for taking time and efforts to review our manuscript. We also appreciate your constructive suggestions and comments, to which we completely agree, and have made extensive revisions to our manuscript accordingly. The point-to-point responses to your comments and concerns are listed as follows:

Reviewer #1 (Remarks to the Author):

This is a timely, interesting and important manuscript that reports significant and new findings implicating involvement of Cochlin and SFRP1 towards development of myopia. Manuscript, despite providing strong evidence in support of reported findings/conclusions, necessitates revision for further serious consideration.

1) Abstract does not report lens induced myopia (LIMS) model. Authors should also validate a subset of data using primary RPE cells and abstract need to incorporate the same.

Thank you very much for pointing this out. We have added the information of LIM model in the Abstract. Please see line 11-12. We also validated the gene expression data using human primary RPE cells, and this information has been added to the Abstract. Please see line 16-18.

2) Abbreviations should be spelled out at their first appearance throughout the text.

Thank you very much. We corrected this problem, ensuring that the full name was used when the abbreviation first appeared in the manuscript.

3) Page 23, line 575, 577, What is diluent? The composition of diluent needs to be described.

The diluent was neutral RIPA buffer. Please see line 593-596. The composition of the diluent was described in line 583-585.

4) Page 24, line 612, please write make and model of mass spec.

The make and model of mass spec was a Thermo Scientific Q Exactive LC-MS/MS system. Please see line 631-632.

5) Page 24, line 615, ProteinPilot 4.0 is not offered by Thermo. Authors need to correct this information.

Thank you very much for pointing this out. This mistake has been corrected. Please see line 634.

6) Page 27, line 693, did they use any other RPE cells? It will be important to use some primary RPE cells (preferably human) to validate at least a subset of experiments.

This is a very good suggestion. We followed your suggestion and validated the expression of *SFRP-1* gene at both transcription and protein levels using quantitative realtime PCR and ELISA, respectively, in human primary RPE cells. Please see Figure 4J and K and line 234-242 in the Results.

7) Please review the reference style including font for formatting conformity.

The reference style, including the font and formatting, has been carefully checked.

8) Page 44, table 1, MAP1B value is 0.000, either it should be eliminated, or a real value be included.

A real value was included.

9) Page 48, Figure 1, Perhaps authors can consider including a cross section of eye cartoon synthesizing measurements 1A-E and pictorially depicting what will myopia look like with respect to measurements in a cross-sectional view? This can be included in current void space in the middle of the figure panels.

Thank you for your suggestion. The cartoon was added as Figure 1H.

10) Page 49, Figure 2B, can they replace IDs with gene names? or include gene names side by side?

The gene names have replaced the IDs. Please see the current Figure 2B.

11) Page 50, Figure 3 A-C, please consider labeling panel A with a different color so that it is clear to the readership.

Thank you for your suggestion. The labeling was put on the left side of the immunohistochemical figures. Please see Figure 3O.

12) Page 50, Figure 3, Panel G and throughout for Western blot-can authors please

provide the whole blot as supplemental figures? (WB whole blots)

Yes, the whole Western blots for the representative blots of this study were all provided as Supplementary figures.

13) Page 53, Figure 7, Please consider providing a cartoon of eye cross section indicating which location of the retina has been presented in G and H. This can be placed adjacent to empty space next to Figure H.

The cartoon was added as Figure 7I.

14) Page 55, Figure 8, The summary cartoon should have some rethink. For example, the blurred vision should perhaps be at the end after the arrow of Non pathologic myopia.

We thought carefully about this question. However, we think the blurred vision should be the start point of the whole signaling axis leading to pathogenesis of nonpathologic myopia. Figure 8 illustrated the model we proposed after summarizing the experiment results in this study. Please see line 400-426 in the Discussion for the detailed description of the formation and proof of the hypothesis in this study.

In summary, the manuscript needs revisions, light English editing for brevity and better presentation.

Thank you for your suggestion. We carefully revised the language and presentation of this manuscript. In addition, we used the paid online editing service of American Journal Experts to polish the English language and grammar of our manuscript. We also invited a native English-speaking collaborator to proofread our manuscript.

Authors should consider using primary RPE for validation and provide Western blots as supplements for review of the readership.

These have been done in the revision. Please see Figure 4J and K for primary human RPE cell validation, and see Supplementary figures for the whole Western blots.

The findings of molecular links to myopia are significant advancements which is likely to exert wide impact in ophthalmology and vision science as well our understanding of cell/tissue signaling.

Thank you very much for the comments.

Reviewer #2 (Remarks to the Author):

This manuscript addresses the molecular mechanisms that may underlie myopia. The authors use the guinea pig to model form-deprived myopia (FDM) and lens-induced myopia (LIM) and test the possible contribution of scleral hypoxia as a cause of myopia. Through proteomic studies the authors identify colchlin, an ECM component, as a protein up-regulated in both models. Using different approaches, they show that the protein localises to the retina and perhaps the RPE. Using a RPE derived cell line, ARPE-19, they show that colchlin addition promotes the up-regulation of the secreted protein SFRP1, which, in turn, stimulates CaMKII in RF/6A, a cell line derived from choroid vasculature. They conclude that a so called “axis” composed of cochlin in the retina, SFRP1 in the RPE, and CaMKII in choroidal vascular endothelial cells is responsible for non-pathologic myopia and suggest that cochlin and SFRP1 are potential therapeutic targets for myopia.

This is a potentially interesting study. However, the overall rationale of the study is difficult to follow and the logic followed by the authors is difficult to grasp especially for non-specialists.

In a revised manuscript, we tried to explain in detail the rationales under the key experiments, which will be mentioned one-by-one through addressing the reviewer's concerns and comments.

The conclusions are largely overstated especially because some of the critical experiments are based on the use of cell lines that are far from representing the in vivo situation. In addition, there are factual mistakes that should be corrected. Overall the authors should re-elaborate their manuscript to facilitate understanding the logic they followed.

Thank you very much for pointing this out. We paid close attention to toning down the conclusion and explaining the rationale underlying the key experiments in detail.

For example, some of the information provided in the discussion would be better placed in the introduction (e.g. SFRP1 has been associated to myopia in GWAS studies)

Thank you for pointing this out. The information on GWAS studies for human *SFRP-1* gene was placed in the Introduction, please see line 92-95.

or to explain why the picked one gene/protein vs any other.

The reasons for selecting cochlin and *SFRP-1* genes as the potential targets were explained in detail in the revised manuscript. Please see line 165-172 in the Results for cochlin gene; line 230-234 in the Results for *SFRP-1* gene.

The conclusions should be toned down as they study simply show correlations. The up-regulation of the indicated “axis” may simply be a consequence and not a cause to myopia.

Yes, we mentioned this in the study limitations in the Discussion. Please see line 495-501.

Additional specific comments are listed below:

1. The introduction is rather focused and does not clearly explain what the rationale of the study.

Yes, we agree with you. We re-wrote the Introduction extensively. Please see 68-115.

2. There is little information on how the guinea pig form of deprived myopia (FDM) and lens-induced myopia (LIM) were established. The related methods are copy paste of those described in ref 44 and these in turn come from an additional reference of the authors from 2011. This information is critical to reproduce the study and it should be described better.

Thank you very much for your suggestion. The methods for establishing FDM and LIM guinea pig models were revised and described in detail in Line 523-535.

3. Line 157. The authors mention that cochlin gene expression is up-regulated in both FDM and LIM. Besides the fact that they have mostly looked at the protein, there is need to explain what the gene encodes for and why they have focused on this protein

in particular vs all the others they found up or down regulated in both models.

This information was added in line 78-79 in the Introduction, line 166-172 in the Results.

It will be also important to understand how this up-regulation occurs in the LIM model

The cochlin expression at both mRNA and protein levels in LIM models was shown in Figure 3K, L, N, O, and P. Please see line 198-211.

4. Table 1, the name of the genes and the name of the proteins are mixed up. For example, the name of the colchlin gene is COCH whereas the protein is called colchlin. Similarly, the KRT19 gene encode keratin 19...etc. The table needs to be corrected

Table 1 has been corrected.

5. Line 202. The authors suggest that Cochlin may be “an early interventional target for non-pathologic myopia”. At this point of the study this is an overstatement. There is only a correlation between Cochlin over-expression and myopia. No evidence that this is the cause of myopia.

Thank you very much for your suggestion. The sentence has been deleted. Please see line 212-215.

6. It is totally unclear why the authors have decided to stimulate the ARPE-19 (and not APE-19 as indicated in line 207) cells with cochlin. What is the rationale behind? This should be explained.

This is a very good question. We explained the rationale behind stimulating ARPE-19 cells with cochlin in the Introduction. Please see Line 73-87.

The ISH distribution does not support a localisation in the RPE, as indicated by the authors, as only minimal staining is observed in this layer. On the contrary, the protein seems to localise in the outer nuclear layer (among photoreceptor cell bodies) and perhaps in the initial portion of the outer segment.

Yes, we agree with you. We revised the manuscript and changed the description of cochlin localization. Please see line 206-209 in the Results.

Why not testing the effect on photoreceptors?

This is a very good question. Based on the literature, cochlin is a secretory protein,

and has been reported to function as an ECM both in the inner ear and in the trabecular meshwork of the eye. Moreover, our IHC results were essentially consistent with the role of cochlin as an ECM at the interface between photoreceptor cells and the RPE. Our hypothesis is that cochlin could serve as a retinal molecular cue to interact with RPE and relay the myopigenic information to RPE, therefore we were more interested in finding the possible interaction between cochlin and RPE. Certainly, the possibility that cochlin may exert some effects on retinal photoreceptors could not be excluded and would be interesting to explore in the future.

Furthermore, ARPE-19 cells assume some RPE characteristics only if properly stimulated. It is unclear if this has been done, no information is available in the methods. The authors cannot use ARPE-19 and RPE interchangeably because this is not the case.

Thank you very much for your reminder. We were careful about this in the revised manuscript, for example, we stated “cochlin-stimulated ARPE-19 cells” instead of “cochlin-stimulated RPE cells”. In addition, we validated the expression patterns of *SFRP-1* gene both at transcript and protein levels in human primary RPE cells, please see Figure 4J and K, and line 237-245 in the Results.

7. Again, why did the authors focus on SFRP1 and not in any of the other genes that were up-regulated upon APRE-19 treatment with cochlin? Perhaps indicating that SFRP1 has been associated with myopia in GWAS studies will help. Addressing the distribution of SFRP1 will also be relevant here.

Thank you very much for your question. We explained the rationale behind selecting SFRP-1 as another target in line 87-100 in the Introduction and in line 232-236 in the Results.

8. What is the rationale of incubating RF/6A cells with the condition media of cochlin treated APRE-19 cells? There is no explanation.

The rationale was explained in line 95-100 in the Introduction. Two sets of experiments were performed in this study, one set using human SFRP-1 recombinant protein

(Figure 5) to examine the direct effects of SFRP-1; the other using SFRP-1-containing conditioned media (Supplementary figure 2) to underscore that the signal was of a RPE origin.

9. SFRP1 is commercially available. The authors need to treat the cells with SFRP1 to demonstrate the possibility that this protein induces apoptosis of RF/6A cells. All experiments with conditioned media are indirect. Furthermore, why focusing on apoptosis and tubulogenesis?

Yes, we examined the effects of SFRP-1 recombinant protein on proliferation, apoptosis, tube formation, and migration of RF/6A cells, which are the basic functions of vascular endothelial cells. Please see Figure 5, and line 246-295 in the Results. In the choroidal vascular endothelial cells, SFRP-1 could cause promoted apoptosis, weakened tube formation, and enhanced migration, which were dysfunctions of vessel endothelial cells. We did not focus on apoptosis and tube formation in particular.

10. What is the Matrigel assay? This should be explained to allow non-experts to follow the logic of the study.

In the Matrigel assay, a hydrogel containing ECM proteins and growth factors is placed into the culture wells and allowed to solidify. As its name indicates, the solidified gel serves as the basement membrane and ECM for vascular endothelial cells. The endothelial cells form tube-like structures after being seeded on the gel and cultured in the incubators for 8 hours. This assay has been widely used for evaluating the tubulogenic activity of endothelial cells. By the way, Transwell assay is used to assess the migratory activity of endothelial cells. If the endothelial cells are dying (promoted apoptosis), actively move away from the vessel framework (enhanced migration), and the disassembling vessel framework cannot be supplemented and strengthened by the endothelial cells (weakened tube formation), then the vessel framework is dismantling, the vessel density is reduced, and the blood perfusion to this tissue is waned. Therefore, vascular endothelial cell dysfunctions may lead to reduced vascular density and blood perfusion.

11. Line 276. Again, the authors overstate the value of their results. The aggregates shown in figure 5D cannot be considered as “tubules” nor what observed in RF/6A cells can be translated to the choroid vasculature as such.

Yes, we agree to your opinion. Figure 5 has been replaced by examining direct effects of SFRP-1 on RF/6A cells, which we think is more obvious than those of SFRP-1-containing conditioned media. Please see the representative pictures for both Matrigel assay (Figure 5D) and Transwell assay (Figure 5F), the effects were more significant. The figure describing the effects of the conditioned media has been placed in Supplementary figure 2.

As for the RF/6A cells, we agree that they cannot be translated to the choroidal vasculature. This is why we have Figure 6, in which we used OCTA to measure choroidal blood vessel perfusion in live animal model and used HE staining and morphometry to analyze the area of choroidal blood vessel lumen.

12. Line 287 The authors consider that canonical Wnt signalling was activated in the RF/6A cells upon exposure to Wnt3a. This seems an erroneous conclusion. The phosphorylated form of beta-catenin increases when the pathway is inactive and thus the protein is targeted for degradation.

Thank you very much for pointing this out. Yes, that was a mistake. When canonical Wnt signaling is activated by Wnt3a, the levels of phosphorylated beta-catenin should be reduced and those of non-phosphorylated beta-catenin should be increased. In the revised manuscript, we ordered an antibody to non-phosphorylated beta-catenin, please see Table 3, and indeed, the levels of non-phosphorylated beta-catenin were increased by Wnt3a. Please see Figure 5 I and J, and line 302-314 in the Results.

13. The data supporting CaMKII activation in the presence of the condition medium are unclear. The biological properties of CaMKII are regulated by multi-site phosphorylation. What should be the mechanism? Does addition of SFRP1 alone mimic CaMKII hypothetical activation?

The current Figure 5 describes the direct effects of SFRP-1. We added an experiment

showing that SFRP-1 could increase the intracellular concentration of calcium, which in turn could activate the downstream phosphorylated CaMKII in RF/6A cells. Please see Figure 5H, I, and K, and line 315-321 in the Results. As for the mechanism about how SFRP-1 could increase the intracellular calcium concentration, we discussed the possible reasons in the Discussion, please see line 461-475 in the Discussion.

How does CaMKII become over-expressed? SFRP1 has been shown to act as a multifunctional modulator of Wnt signalling as well as to regulate other pathways (Notch) as well as to interact with other proteins. This should be mentioned and considered at least in the discussion.

In the revised manuscript, following confirmation of repeated experiments, we found that the levels of total CaMKII did not vary significantly among the experiment cell groups (Figure 5I and M; line 321-325), however, its levels changed significantly in the in vivo experiments (Figure 7E, G, H, J; line 393-401). We discussed the possible reason in the Discussion, please see line 475-483.

Reviewer #3 (Remarks to the Author):

The authors conducted a comprehensive experiment and try to figure out the role of cochlin/SFRP1/CaMKII axis in FDM/LIM development. At first, proteomic analysis of ocular posterior pole had identified that cochlin was the strongest DEG. Then the time dependent expression patterns of cochlin were validated by a series of experiments and the RPE was found to be expressing this protein; By further stimulating the RPE with cochlin, SFRP1 was found to be highly expressed. The authors also demonstrated how cochlin and SFRP1 influence the choroidal vascular remodeling and their effect on FDM formation. These findings are important to unveil the biological mechanism of myopia development, especially important to figure out the 'RPE and choroid mechanisms' in myopia development.

Overall, this study is well designed, with clear logic, proper experimental techniques applied and sufficient in vivo and in vitro evidence. Meanwhile, the manuscript is well

written, the figures are clear and well organized.

Thank you for the comments.

Concerns:

Cochlin is a secretory ECM protein and was found in glaucomatous trabecular meshwork, but its role in myopia development is unclear. Although this study is driven by omic analysis, which is hypothesis free in the initial stage, a clear hypothesis should be formed for following analysis.

Thank you very much for your suggestion. A hypothesis has been put forward in the Introduction, please see line 68-109.

Whether cochlin is a consequence induced by axial elongation stress or it is caused by blur vision remain unclear.

Yes. We stated this as the study limitations in the Discussion. Please see line 497-503.

More importantly, the mechanism that how cochlin is interplay with SFRP1 and CaMKII and the potential signaling pathway is not specified. It will be good to have a theoretical discussion and mention the limitation of this study in the discussion section.

We streamlined the formation and proof of our hypothesis in line 410-426 in the Discussion. The possible interaction between cochlin, RPE, and SFRP-1 was discussed based on the literature in line 450-454. The possible mechanism underlying the SFRP-1-mediated activation of noncanonical Wnt/Ca²⁺/CaMKII was discussed in line 466-475.

Minor:

1. In the introduction section, it will be nice to include background knowledge of cochlin and the hypothesis of cochlin and myopia development.

This is a good suggestion. Please see line 68-109 in the Introduction for the background knowledge of cochlin and the hypothesis.

2. In the section of 'Validation of Cochlin expression in retinas of FDM and LIM models': From line 179-185, authors explained the difference of cochlin protein expression levels in the outer segments of the ocular at 6w time points, however, in the coming

section (from line 196), authors only demonstrated it's expression pattern of the first 3 weeks. It will be more clear if the results were displayed in a chronological order, e.g the 1) validation of the expression at early stages; 2) late stages; 3) location of cochlin protein expression;

Thank you very much for your suggestion. We have revised the figure according to your suggestion, please see Figure 3, and line 186-217 in the Results.

3. Figure 3M, please shift the x-axis to the same scale of Figure 3N if they the measurements are taken from the same samples.

This scale on the x-axis has been adjusted. Please the current Figure 3D-I.

4.Line 223: Miss spell 'SPRP1'.

Thank you. It has been corrected.

The above is a brief summary of our revision. We sincerely wish that the current manuscript could meet the requirements for publication in your journal. Please feel free to contact us If there are further questions.

Corresponding author

Yan Zhang

Tianjin Medical University Eye Hospital

April 29th, 2023

Reviewers' comments:

Reviewer #1 (Remarks to the Author):

The authors have addressed all review comments raised during the last round of review.

Reviewer #2 (Remarks to the Author):

The revised version of the manuscript is improved. Nevertheless, there are a number of points that still need attention.

1. Images in Fig 3O do not support the proposed increased expression what it is not clear what it has been quantified. Immunofluorescence should be used to perform this analysis. As the fluorescent signal can be better quantified
2. In line 274 the authors indicate that cells undergo "Tubulation". This cannot be appreciated in the images. Cells simply are aggregated. There is no tubule formation. This needs to be corrected or images of tubules should be shown
3. The data reporting beta-catenin and CAMkII activity are still very confusing. The authors should report the ration of phosphorylated vs total levels of the proteins. It is unclear what is the difference between non-p- beta catenin vs beta catenin? What are the antibodies used in each case? The effect of Wnt3a alone should be tested
4. Fig 7: In all westerns, the GAPDH signal is oversaturated. This makes it difficult to quantify the bands. Less exposed westerns should be used for quantification. Furthermore, CamKII should be expressed as the ration between phosphorylated/over total levels
5. In line 107: "secretion, with in turn might" should read "secretion, which in turn might"
6. Line 326: The sentence "Taken together with the results of the functional assays, these results imply that..". Would read better as follow "The results of the functional assays imply that"
7. Although the text reads better, there are still some difficult sentences and typos that should be corrected

Reviewer #3 (Remarks to the Author):

The authors have address the questions mentioned in previous revision successfully. However, the language is not as clear as the previous version, and professional paragraphing by native English speakers is strongly recommended. Several suggestions for the authors.

- 1.Line 16, difficult to understand: "Next, microarray- and quantitative PCR-based analyses of cochlin-stimulated human RPE cell line and primary cells revealed increased expression of secreted frizzled-related protein 1 (SFRP1), which elevated intracellular Ca²⁺ concentration, activated Ca²⁺/calmodulin-dependent protein kinase II (CaMKII) and caused dysfunctions in choroidal vascular endothelial cells, leading to reduced blood perfusion in the choroid and consequent nonpathologic myopia."
- 2.Line23, difficult to understand: "Collectively, this study identified through comprehensive omics analyses a novel signaling axis constituted by cochlin in the retina, SFRP1 in the RPE, and CaMKII in choroidal vascular endothelial cells and responsible for the incidence of nonpathologic myopia,

implicating cochlin and SFRP1 as potential targets for myopia control. "

3.Line 50: I don't think the "wearing spectacles or orthokeratology lens , administering low-dose atropine" indicates the "vague understanding of pathogenic mechanism". These classical interventions has been supported by theoretical and experimental studies ...Please make clear and humble statements.

4.Line 53 : Space before a paragraph? Please keep consistence.

5.Line 53-67: A little bit redundant and not relevant to your theory. Please be more concise.

6.Line 68-116: This paragraph is too much and difficult to follow...Please can you summarize you research hypothesis? If you have a targeted molecule or pathway that is associated with choroidal circulation, please be clear which part has been validated and what is the research gap, then what you would like to do to fill this research gap. If this is a hypothesis-free study, please can you explain why you would like to use proteomics and what following steps you have conducted to validate your findings from omic-analysis.

7.Line 122: How this proportion was calculated? Why with longer treatment, the decreased proportions are smaller?

8.Line 164: According to Figer 2B, COCH was downregulated in the comparison between LIM vs. FDM. It is suggested to run a comparison of FDM vs. LIM, which gives a better indication that the more myopic eye contains higher expression of Coch.

In general, this study is carefully designed and conducted, with rich evidence to support the role of cochlin/SFRP1/CaMKII axis in myopia development. However, the language should be improved to make it more clear to other readers.

Dear reviewers:

Thank you very much for taking the time and effort to review our manuscript. We completely agree with your suggestions and comments, based on which we have extensively revised our manuscript. The point-by-point responses to your comments and suggestions are listed as follows.

Reviewers' comments:

Reviewer #1 (Remarks to the Author):

The authors have addressed all review comments raised during the last round of review.

Thank you very much.

Reviewer #2 (Remarks to the Author):

The revised version of the manuscript improved. Nevertheless, there are a number of points that still need attention.

1. Images in Fig 3O do not support the proposed increased expression what it is not clear what it has been quantified. Immunofluorescence should be used to perform this analysis. As the fluorescent signal can be better quantified.

Thank you very much for pointing this out. We performed additional immunohistochemistry (IHC) experiments and changed the representative pictures, which we think can better represent the relative expression levels of the cochlin protein among the experimental groups. Please see Figure 3M in the current manuscript. We used ImageJ to quantify the IHC pictures. Briefly, we converted the color images into binary black and white images. Then, we set up an upper limit and a lower limit, ensuring that only positive staining signals, i.e., the brown signals in the photoreceptor layer and outer nuclear layer were selected. Then, the staining intensity of the brown signals was calculated by ImageJ and normalized to the area of the retina (μm^2). We term the normalized staining intensity the “mean staining intensity”. We retained the IHC results in Figure 3 because we think these results showed the delicate structure of the retina and the clear localization of the cochlin

staining signal. Please see the description of the results in lines 192-201.

We also performed immunofluorescence. To validate the localization of the cochlin protein in myopia retinas, rhodopsin and cochlin double immunofluorescence staining was performed. Briefly, following deparaffinization, rehydration, heat-mediated antigen retrieval, and blockage, mouse monoclonal anti-rhodopsin (1:1000) and rabbit polyclonal anti-cochlin (1:1000) primary antibodies were incubated with the sections at 4 °C overnight. After extensive washes, Alexa 647-conjugated goat anti-mouse (1:2000) and Alexa 488-conjugated goat anti-rabbit (1:2000) antibodies were incubated with the sections for 2 h at room temperature. The pictures were taken under a confocal microscope (Zeiss, Germany). The results are as follows.

Figure 1. Immunofluorescence staining of cochlin in the retinas of guinea pig myopia models. A representative picture of double immunostaining is shown in (A). The green fluorescence signal (cochlin) was primarily localized at the outer

nuclear layer and the inner segments of retinal photoreceptors, with sparse distribution in the outer segments. The red fluorescence signal (rhodopsin) was localized at the outer segments of retinal photoreceptors, with dispersed colocalization with the green signal (cochlin). The localization of the cochlin fluorescence signal is consistent with that shown by immunohistochemistry (brown signal) in Figure 3M. Representative pictures of cochlin immunofluorescence staining in normal and myopic retinas are shown in (B). The intensity of the green fluorescence was quantified using ImageJ and normalized to the area of the retina, which was termed the “mean fluorescence intensity” (C). The trend of mean fluorescence intensity among the experimental groups is consistent with that shown by IHC in Figure 3N.

2. In line 274 the authors indicate that cells undergo “Tubulation”. This cannot be appreciated in the images. Cells simply are aggregated. There is no tubule formation. This needs to be corrected or images of tubules should be shown.

Thank you very much for your suggestion. We have changed this. Please see lines 265-266 and lines 285, 418, 840, and 841-843.

3. The data reporting beta-catenin and CAMKII activity are still very confusing. The authors should report the ration of phosphorylated vs total levels of the proteins. It is unclear what is the difference between non-p- beta catenin vs beta catenin? What are the antibodies used in each case? The effect of Wnt3a alone should be tested

For β -catenin, we searched the literature and found that β -catenin has been generally recognized as one of the surrogates of canonical Wnt signaling activation (PMID: 37042252; 36522719; 33176855; 33153498), i.e., high levels of β -catenin indicate activation of canonical Wnt signaling, whereas low levels of β -catenin indicate inactivation. Therefore, we decided to use western blots to detect β -catenin levels in the experimental cell groups, and the results are shown in Figure 5I and J. The antibody used for detecting β -catenin is listed in Table 3.

For CaMKII activity, we show the ratio of phosphorylated CaMKII to total CaMKII (p-CaMKII/CaMKII) in Figure 3 (for cell cultures) and Figure 7 (for the FDM model). Please see lines 306-317 and lines 385-396 for the result description and lines 470-480 for the discussion.

We also detected the effects of Wnt-3a alone on canonical Wnt signaling, and the results are as follows.

Figure 2. The effects on Wnt-3a on canonical Wnt signaling. Recombinant human Wnt-3a (0.3 μ g/ml, 99% homology to monkey Wnt-3a) and mouse Wnt-3a (0.2 μ g/ml, 97% homology to monkey Wnt-3a) were incubated with monkey choroidal vascular endothelial cells RF-6A for 24 h (n=3), and the β -catenin protein levels were detected by western blots (A). The western blots were quantified using ImageJ (B).

This may serve as a positive control for our experiments. As shown by the western blots, both human and mouse recombinant Wnt-3a dramatically elevated the protein levels of β -catenin, indicating that the canonical Wnt signaling pathway in RF-6A cells was intact and could be activated by Wnt-3a, an activator of the canonical Wnt signaling pathway.

4. Fig 7: In all westerns, the GAPDH signal is oversaturated. This makes it difficult to quantify the bands. Less exposed westerns should be used for quantification. Furthermore, CamKII should be expressed as the ration between

phosphorylated/over total levels.

Thank you very much for your suggestion. We used less exposed GAPDH signals for representative pictures and for western blot qualification. Please see Figure 3B and K, Figure 5I, Figure 7A, C, E, I, Supplementary figure 1, 2, 4, 5, and 7.

For CaMKII activity, we show the ratio of phosphorylated CaMKII to total CaMKII (p-CaMKII/CaMKII) in Figure 3 (for cell cultures) and Figure 7 (for the FDM model). Please see lines 306-317 and lines 385-396 for the result description and lines 470-480 for the discussion.

5. In line 107: “secretion, with in turn might” should read “secretion, which in turn might”

Thank you very much for your suggestion. We have rephrased the sentence; please see lines 87-92.

6. Line 326: The sentence “Taken together with the results of the functional assays, these results imply that..”. Would read better as follow “The results of the functional assays imply that”

Thank you very much for your suggestion. We have rephrased the sentence. Please see lines 318-321.

7. Although the text reads better, there are still some difficult sentences and typos that should be corrected.

Thank you for your comments. We proofread the manuscript several times and invited American Journal Experts to polish the manuscript.

Reviewer #3 (Remarks to the Author):

The authors have address the questions mentioned in previous revision successfully. However, the language is not as clear as the previous version, and professional paragraphing by native English speakers is strongly recommended. Several suggestions for the authors.

Thank you very much for your comments. We invited American Journal Experts to polish the manuscript. We hope that the readability and clarity of this version of the

manuscript have been improved.

1. Line 16, difficult to understand: “Next, microarray- and quantitative PCR-based analyses of cochlin-stimulated human RPE cell line and primary cells revealed increased expression of secreted frizzled- related protein 1 (SFRP1), which elevated intracellular Ca²⁺ concentration, activated Ca²⁺/calmodulin-dependent protein kinase II (CaMKII) and caused dysfunctions in choroidal vascular endothelial cells, leading to reduced blood perfusion in the choroid and consequent nonpathologic myopia.”

Thank you very much for your comments. We have revised the sentences; please see lines 16-21.

2. Line23, difficult to understand: “Collectively, this study identified through comprehensive omics analyses a novel signaling axis constituted by cochlin in the retina, SFRP1 in the RPE, and CaMKII in choroidal vascular endothelial cells and responsible for the incidence of nonpathologic myopia, implicating cochlin and SFRP1 as potential targets for myopia control. “

Thank you very much for your comments. We have rephrased the sentences; please see lines 23-27.

3. Line 50: I don't think the “wearing spectacles or orthokeratology lens , administering low-dose atropine” indicates the “vague understanding of pathogenic mechanism”. These classical interventions has been supported by theoretical and experimental studies ...Please make clear and humble statements.

We completely agree with you. Thank you very much for pointing this out. We have revised the sentences. Please see lines 47-51.

4. Line 53: Space before a paragraph? Please keep consistence.

Thank you very much. We have corrected the formality inconsistency.

5. Line 53-67: A little bit redundant and not relevant to your theory. Please be more concise.

Thank you for your suggestion. We have rewritten that paragraph. Please see lines 52-72.

6. Line 68-116: This paragraph is too much and difficult to follow...Please can you summarize your research hypothesis? If you have a targeted molecule or pathway that is associated with choroidal circulation, please be clear which part has been validated and what is the research gap, then what you would like to do to fill this research gap. If this is a hypothesis-free study, please can you explain why you would like to use proteomics and what following steps you have conducted to validate your findings from omic-analysis.

Thank you for your suggestion. We propose a hypothesis based on our preliminary results and the literature. We have rewritten the paragraph; please see lines 73-92.

7. Line 122: How this proportion was calculated? Why with longer treatment, the decreased proportions are smaller?

This is a very good point. Thank you very much. We checked the data and corrected the calculation error. Please see lines 102-105.

8. Line 164: According to Figer 2B, COCH was downregulated in the comparison between LIM vs. FDM. It is suggested to run a comparison of FDM vs. LIM, which gives a better indication that the more myopic eye contains higher expression of Coch.

Yes, we agree with you. We follow your suggestion. We plotted the gene expression of cochlin in the normal control, LIM, and FDM groups together. Please see Figure 3J. We also ran western blotting with the samples from the 3 groups together. Please see Figure 3K and L. Please also see lines 182-191 for description of the results.

In general, this study is carefully designed and conducted, with rich evidence to support the role of cochlin/SFRP1/CaMKII axis in myopia development. However, the language should be improved to make it more clear to other readers.

Thank you very much for your comments. We proofread and revised the

manuscript several times and invited American Journal Experts to polish the manuscript.

The above is a brief summary of our revision. We sincerely hope that the current manuscript meets the requirements for publication in your journal. Please feel free to contact us if there are further questions.

Corresponding author

Yan Zhang

Tianjin Medical University Eye Hospital

July 4th, 2023

REVIEWERS' COMMENTS:

Reviewer #2 (Remarks to the Author):

The revised version of this manuscript is improved at least in term of presentation.

Still there are conceptual mistakes that persists related to Wnt canonical signalling activation. The authors use the protein levels of beta-catenin as a measure of pathway activation. This is wrong, independently of the few publications that the authors cite to support this decision. The activation occurs by phosphorylation and the authors should measure the ratio between total beta-catenin and phopho-beta-catenin. Alternatively, they could measure the level of expression of any target of the pathway, e.g. axin-2. It also very strange that addition of wnt3a and Sfrp1 lead to pathway activation as Sfrp1 should antagonize Wnt3a (even though the amount of Sfrp1 used is 10-fold less than that of Wnt3a, that, surprisingly, has not been tested alone)

Furthermore, the pattern of staining shown in Figure 1 of the rebuttal letter is very different (and cannot be appreciated) from that shown in Figure 3. The difference should be clarified and the immunofluorescence staining should be presented at least as a supplementary figure.

Reviewer #3 (Remarks to the Author):

The authors have address the questions mentioned in previous revision successfully. The language is improved and the overall logic is much more clear.

Dear reviewers:

Thank you very much for your time and effort to review our manuscript. We completely agree with your suggestions and comments, based on which we have revised our manuscript. The point-by-point responses to your comments and suggestions are listed as follows.

REVIEWERS' COMMENTS:

Reviewer #2 (Remarks to the Author):

The revised version of this manuscript is improved at least in term of presentation.

Still there are conceptual mistakes that persists related to Wnt canonical signalling activation. The authors use the protein levels of beta-catenin as a measure of pathway activation. This is wrong, independently of the few publications that the authors cite to support this decision. The activation occurs by phosphorylation and the authors should measure the ratio between total beta-catenin and phopho-beta-catenin. Alternatively, they could measure the level of expression of any target of the pathway, e.g. axin-2.

Thank you very much for your suggestion. We measured the ratio of phospho- β -catenin/ β -catenin, please see Figure 5h-l, and please see line 299-324 in the Results.

It also very strange that addition of wnt3a and Sfrp1 lead to pathway activation as Sfrp1 should antagonize Wnt3a (even though the amount of Sfrp1 used is 10-fold less than that of Wnt3a, that, surprisingly, has not been tested alone)

Yes, exactly. We expected that the 10-fold more Wnt3a could override the effects of SFRP-1 on the vascular endothelial cells. It turned out to be the case, as revealed by the results of the functional assays, including caspase activity, Matrigel gel, and Transwell assays shown in Figure 5c, d, f, respectively. On the other hand, although the 10-fold less SFRP-1 may antagonize Wnt3a a little, the amount of Wnt3a is too much, so the Wnt3a can still activate the canonical Wnt signaling. The relative actional intensity between SFRP-1 and Wnt3a would be interesting to test in the cell culture in our future study.

Furthermore, the pattern of staining shown in Figure 1 of the rebuttal letter is very different (and cannot be appreciated) from that shown in Figure 3. The difference should be clarified and the immunofluorescence staining should be presented at least as a supplementary figure.

Yes. Thank you for your suggestion. We include the immunofluorescence staining of cochlin as Supplementary Fig. 2. We also mention the different staining pattern in the Results, please see line 212-215.

Reviewer #3 (Remarks to the Author):

The authors have address the questions mentioned in previous revision successfully.

The language is improved and the overall logic is much more clear.

Thank you very much for your comments.

The above is a brief summary of our revision. We sincerely hope that the current version of manuscript meets the requirements for publication in your journal. Please feel free to contact us if there are further questions.

Corresponding author

Yan Zhang

Tianjin Medical University Eye Hospital

Aug 8th, 2023